# Intrinsic enzymatic properties modulate the self-propulsion of micromotors

Xavier Arqué[1], Adrian Romero-Rivera[2], Ferran Feixas[2], Tania Patiño[1], Sílvia Osuna[2,3] & Samuel Sánchez [1,3]

Bio-catalytic micro- and nanomotors self-propel by the enzymatic conversion of substrates into products. Despite the advances in the field, the fundamental aspects underlying enzyme-powered self-propulsion have rarely been studied. In this work, we select four enzymes (urease, acetylcholinesterase, glucose oxidase, and aldolase) to be attached on silica microcapsules and study how their turnover number and conformational dynamics affect the self-propulsion, combining both an experimental and molecular dynamics simulations approach. Urease and acetylcholinesterase, the enzymes with higher catalytic rates, are the only enzymes capable of producing active motion. Molecular dynamics simulations reveal that urease and acetylcholinesterase display the highest degree of flexibility near the active site, which could play a role on the catalytic process. We experimentally assess this hypothesis for urease micromotors through competitive inhibition (acetohydroxamic acid) and increasing enzyme rigidity (β-mercaptoethanol). We conclude that the conformational changes are a precondition of urease catalysis, which is essential to generate self-propulsion.

[1] Institute for Bioengineering of Catalonia (IBEC), The Barcelona Institute of Science and Technology (BIST), Baldiri i Reixac 10-12, 08028 Barcelona, Spain. [2] CompBioLab Group, Institut de Química Computacional i Catàlisi (IQCC) and Departament de Química, Universitat de Girona, Carrer Maria Aurèlia Capmany 69, 17003 Girona, Spain. [3] Institució Catalana de Recerca i Estudis Avançats (ICREA), Pg. Lluís Companys 23, 08010 Barcelona, Spain. Correspondence and requests for materials should be addressed to T.Pño. (email: tpatino@ibecbarcelona.eu) or to S.O. (email: silvia.osuna@udg.edu) or to S.Sán. (email: ssanchez@ibecbarcelona.eu)

Enzymes are biological catalysts that increase the conversion rate from substrates to products. In solution, they can adopt a wide range of conformations which can be crucial to facilitate substrate binding and product release[1,2]. Enzymatic reactions can be divided into two steps: (i) the binding and unbinding of an uncatalyzed molecule to the active site of the enzyme, and (ii) the catalyzed conversion of substrates into products (Supplementary Equations 1 and 2 in Supplementary Note 2). Both steps have been reported to generate enhanced diffusion of single enzymes in solution[3,4]. Several mechanisms have been suggested to produce active motility of single enzymes, including (i) the thermal effect[5] and collective heat discussed by Golestanian[6], both generated from catalyzing exothermic reactions, (ii) the conformational changes arising from catalysis[7-9] or binding–unbinding interactions[10], and (iii) the self-diffusiophoretic mechanism[11]. The contribution of these proposed mechanisms is still under debate and further research is needed to shed more light on the fundamentals of single enzyme active motion[3,10,12,13].

By anchoring enzymes on micro- and nano-sized structures, bio-catalysis can produce a propulsive force that generates self-propulsion of micro- and nanoparticles, named enzymatic micro- and nanomotors. Bio-propulsion represents a more biocompatible and versatile alternative[4,14] to inorganic catalysts engines, which are predominantly operated by toxic fuels, such as hydrogen peroxide[15], and are unsuitable for biomedical applications. The capabilities of bio-catalysis to propel structures in solution were initially reported by Mano and Heller[16], and Feringa and co-workers[17], leading to the further expansion of enzymatically generated active motion[4,12]. Several enzymatic micro- and nanomotors have been reported to use individual enzymes or combinations of different enzymes and inorganic catalysts[4,18-22]. Nanomotors can display enhanced diffusion powered by individual enzymes such as catalase[23,24], urease[14,25,26], glucose oxidase[27], and trypsin[22]. To date, urease[28-31] and catalase[32-36] are the only enzymes that have been reported to individually propel micron-sized structures. Thus, an expansion of the enzyme library for propelling micro- and nanomotors will enable a better understanding of the mechanisms of enzyme-based active motion.

Although a lot of effort has been put into studying the fundamental aspects of the enhanced diffusion of single enzymes, to the best of our knowledge, no experimental investigations have considered the factors ruling self-propulsion of enzyme-powered micro- and nanomotors. There is a lack of understanding on how the catalytic, structural, and dynamic properties of enzymes affect self-propulsion. Moreover, the use of molecular dynamics (MD) simulations has not been reported in this field. Such studies are crucial for the rational design of enzyme-based motors and their effective implementation for on-demand applications.

In this work, hollow silica microcapsules (HSMC) are modified with urease (UR), acetylcholinesterase (AChE), glucose oxidase (GOx), or aldolase (ALS) to study the capacity of each enzyme to power active motion. These 4 enzymes are selected due to their differences in turnover number ($k_{cat}$), to understand the role of catalytic turnover on active motion. Motion dynamics are analyzed through optical microscopy and correlated with two key enzymatic properties: (i) the catalytic conversion rate, experimentally studied, and (ii) the enzyme conformational flexibility, studied by both MD simulations and experimental modulation. We use acetohydroxamic acid (AHA), a reversible competitive UR inhibitor, to study whether the binding and unbinding of AHA to the active site have any effect on (i) catalysis and (ii) enzyme flexibility, and how that affects the active motion. To gain more insight into the role of structural flexibility, we employ different concentrations of β-mercaptoethanol (BME) to increase

the enzyme rigidity near the active site of UR. Our results elucidate the role of conformational changes at a molecular level as a requisite for catalysis and, as a result, on the self-propulsion of enzyme-powered microstructures, shedding light on how the intrinsic enzyme properties govern enzyme-powered active motion.

## Results

**Enzymatic hollow silica micromotors fabrication.** Due to the biocompatibility and ease of surface modification offered by silica[37], HSMC were chosen as the base structure for our micromotors. Figure 1a shows a schematic representation of the synthesis process. Briefly, a silicon dioxide shell was grown onto commercial polystyrene (PS) beads through a modified Stöber method[38]. The polystyrene core was then removed, and the resulting hollow microcapsules were characterized by scanning electron microscopy (SEM) and transmission electron microscopy (TEM). The average diameter of the HSMC was estimated as $2.0 \pm 0.1$ μm from SEM micrographs of 200 particles. Surface roughness and holes can be observed on the capsules in both the TEM and SEM micrographs (Fig. 1b, c). Surface roughness has been reported to promote higher enzyme attachment which is crucial for self-propulsion[29]. We hypothesize that the holes were formed at PS bead contact points during silica shell formation. A false color topographical representation of the TEM imaging clearly shows the opening on the silica capsule (Fig. 1d and Supplementary Fig. 1). The TEM analysis reveals an average number of holes per capsule of $1.2 \pm 0.1$ ($N = 105$). The chemical composition of the resulting HSMC was examined by Fourier-transform infrared spectroscopy (FTIR) confirming silica as the base material (Supplementary Fig. 2 and Supplementary Note 4).

UR, AChE, GOx, and ALS were covalently bound to the silica surface by using glutaraldehyde (GA) as linker obtaining the enzymatic silica micromotors (HSMM) (Fig. 1a). A non-homogeneous distribution on a similar silica surface has been recently reported by our group[29], providing the asymmetric distribution of reaction product required for self-propulsion[11]. We evidenced the presence of protein on the silica surface through Krypton™ fluorescent protein staining (Fig. 1e, Supplementary Fig. 3, and Supplementary Note 5) and quantified the unattached protein using the bicinchoninic acid (BCA) assay (Supplementary Fig. 4 and Supplementary Note 6). Enzymatic activity of the different HSMM was confirmed through detection of the reaction product (Supplementary Figs. 5–8 and Supplementary Notes 7–10) before recording the motion of each micromotor.

**Enzyme-dependent motion dynamics of micromotors.** The self-propulsion of each HSMM was systematically analyzed for different substrate concentrations by testing 20 micromotors per each concentration. Self-propulsion was measured (Fig. 2 and Supplementary Figs. 9–12) by tracking the individual trajectories from which a custom-made software calculated the mean squared displacement (MSD) and extracted the speed (see "Data analysis of motion" in Methods)[39,40].

Only UR and AChE-modified micromotors showed a significant increase in self-propulsion for higher substrate concentration (Fig. 2). When compared to other enzymes, UR clearly produced directional motion (Fig. 2, Supplementary Fig. 9A, and Supplementary Movie 1). By increasing the urea concentration up to 100 mM, the propulsive speed increased to $2.07 \pm 0.25$ μm s$^{-1}$ (Supplementary Fig. 9B). The increase in concentration did not lead to higher speeds but resulted in a slight decrease. This could be due to the relatively high viscosity of the media or the

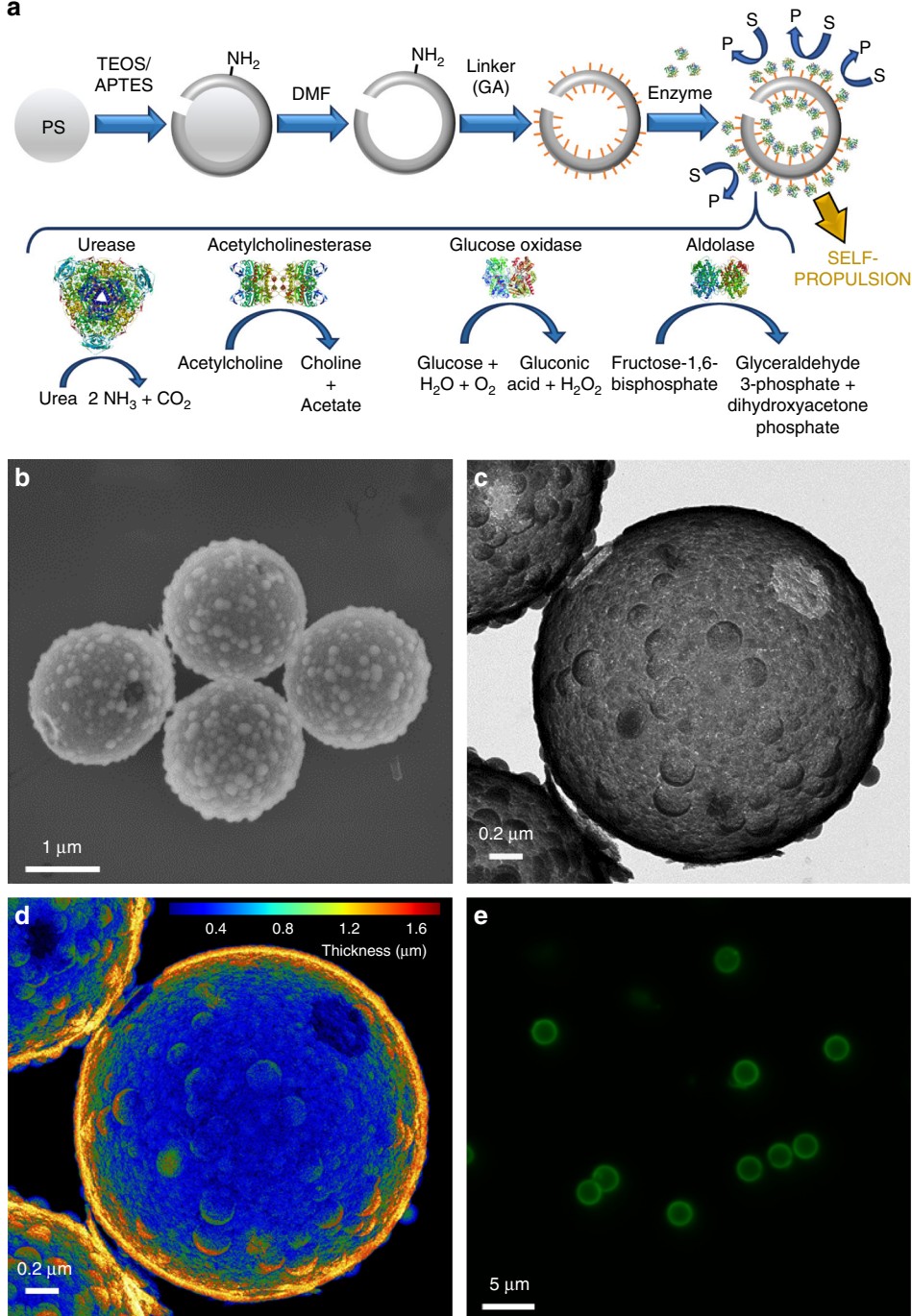

**Fig. 1** Fabrication and characterization of hollow silica micromotors. **a** Schematic representation of the fabrication process of HSMC functionalized with enzymes to obtain HSMM that catalyze substrates (S) into products (P) resulting in self-propulsion. PS: polystyrene, TEOS: tetraethylorthosilicate, APTES: 3-aminopropyltriethoxysilane, DMF: dimethylformamide, GA: glutaraldehyde. Enzyme structures are extracted from RCSB PDB (see Supplementary Note 3). **b** SEM and **c** TEM image of HSMC showing the hole and silica bulks. **d** False color TEM image showing HSMC thickness. **e** Fluorescence (Ex/Em = 520/580 nm) of enzymatic HSMM treated with Krypton™ protein staining dye

inhibition by the substrate, both caused by high urea concentrations.

Compared to UR, AChE micromotors exhibited less directional motion and the speed did not increase as much at acetylcholine (ACh) optimal concentrations (Supplementary Fig. 10A and Supplementary Movie 2). MSD became maximal for concentrations of about 0.1 mM ACh (Fig. 2b), which corresponds to a speed of $0.43 \pm 0.04\ \mu m\ s^{-1}$ (Supplementary Fig. 10B). The speed of AChE-HSMM decreased for higher ACh concentrations,

which could again be attributed to increased viscosity or substrate inhibition[41].

Neither GOx-HSMM nor ALS-HSMM exhibited any active motion for a range of glucose (GLC) and fructose 1,6-bisphosphate (FBP) concentrations, respectively (Fig. 2, Supplementary Figs. 11 and 12, and Supplementary Movies 3 and 4). Both types of micromotors only showed Brownian motion with MSDs and speeds indistinguishable from the corresponding controls.

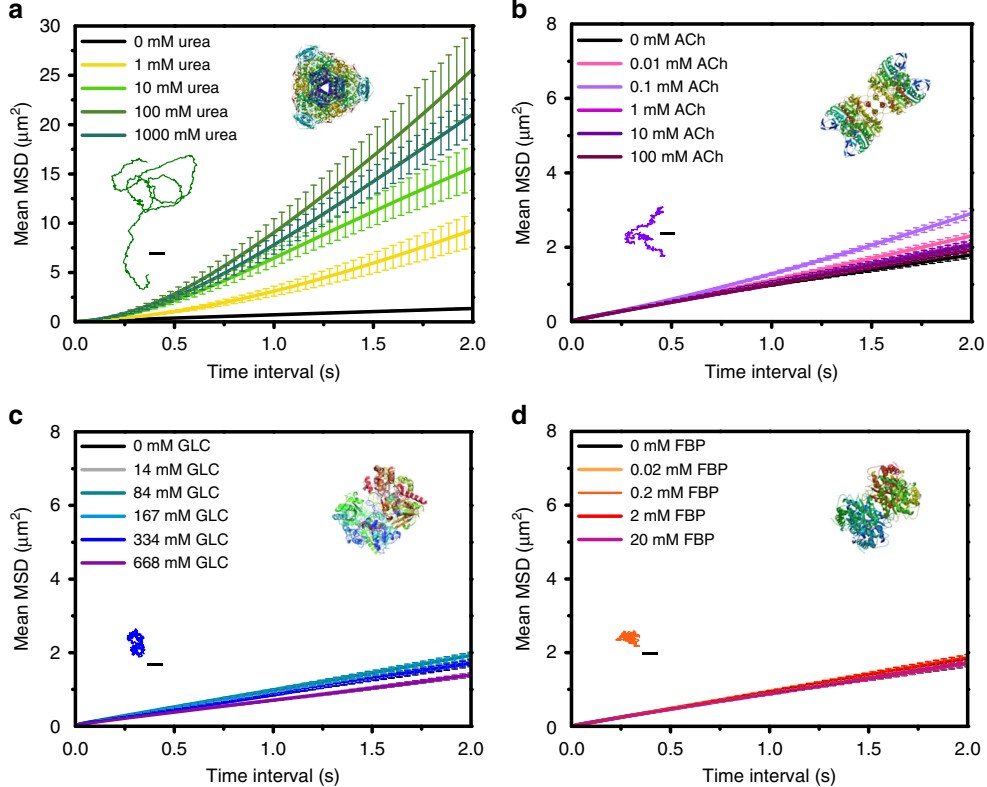

**Fig. 2** Motion dynamics UR, AChE, GOx, and ALS micromotors. Average MSD over time for different substrate concentrations: **a** UR-HSMM, **b** AChE-HSMM, **c** GOx-HSMM, and **d** ALS-HSMM. Inset: representative trajectories for **a** UR-HSMM with 100 mM urea, **b** AChE-HSMM with 0.1 mM acetylcholine (ACh), **c** GOx-HSMM with 334 mM glucose (GLC), and **d** ALS-HSMM with 0.02 mM fructose 1,6-bisphosphate (FBP) (scale bars: 2.5 μm). Enzyme structures are extracted from RCSB PDB (see Supplementary Note 3). Results are shown as the mean ± standard error of the mean (s.e.m.). Twenty particles were analyzed per condition. Source data are provided as a Source Data file

**Comparing intrinsic properties of enzymes**. Among the four tested enzymes, UR produced the strongest self-propulsion of microparticles, followed by AChE, while GOx and ALS did not result in any net self-propulsion (Fig. 3). The four enzymes have different molecular weights and quaternary structures and we observed that the highest size and weight (provided by the purchasing company and extracted from literature) resulted in the highest micromotor's speed, following this increasing order: ALS (150 kDa) and GOx (160 kDa), AChE (230–280 kDa), and UR (440–480 kDa). This difference in size affected the number of enzymes attached to the silica surface obtaining ~38 times more ALS attached on the microparticles than UR (Fig. 3a and Supplementary Fig. 4). Nevertheless, neither increasing the enzyme number for UR micromotors (Supplementary Fig. 13) nor decreasing it for ALS micromotors (Supplementary Fig. 14), yielded to any difference in self-propulsion. In addition, the resulting speed showed a positive correlation with the conversion rate, where the higher the reported turnover number ($k_{cat}$) of an enzyme, the higher the micromotor's speed (Fig. 3b, d): ALS ($k_{cat} = 13\,s^{-1}$)[42], GOx ($k_{cat} = 920\,s^{-1}$)[43], AChE ($k_{cat} = 10,833\,s^{-1}$)[44], and UR ($k_{cat} = 23,400\,s^{-1}$)[25].

Enzymes are dynamic entities undergoing a wide range of conformational changes[1]. This structural dynamism can play a direct role in substrate binding and product release. To identify the most relevant conformational changes occurring in each enzyme, we performed MD simulations and applied the dimensionality reduction technique principal component analysis (PCA) (Fig. 3c, e–h and Supplementary Fig. 15). A higher flexibility near the active site was found in active site loops of UR

(Fig. 3e and Supplementary Movie 5) and AChE (Fig. 3f and Supplementary Movie 6), which exhibited the strongest self-propulsion. This contrasts with the reduced flexibility of GOx (Fig. 3g) and ALS (Fig. 3h), which did not exhibit any capacity for self-propulsion. These results suggest that the conformational changes near the active site of UR and AChE may be directly coupled to catalysis and thus self-propulsion. In ALS and GOx, these changes are far from the active site and have a minor effect along the catalytic cycle.

The studied enzymes also accommodate differently-sized substrates in the active site: urea < acetylcholine < glucose < fructose-1,6-bisphosphate. Therefore, conformational changes associated with the previously highlighted flexible loops in UR and AChE may play a role in regulating the binding of the small urea and acetylcholine substrates, affecting the access to the active site pocket. To further explore the impact of conformational dynamics on substrate access, we analyzed the substrate access tunnels and their bottleneck radius (i.e., the radius of the narrowest part of the tunnel) at the most open and closed conformations observed in the MD simulations (Fig. 3c). This analysis captures how the substrate access to the active site is hindered by the loop flexibility. Interestingly, the conformation of the flap covering the active site in UR significantly modifies the bottleneck (differences of ca. 0.9 Å in the bottleneck radius between closed and open conformations) (Fig. 3e). The same effect is observed in AChE, although to a lesser extent (differences of ca. 0.4 Å) (Fig. 3f). These findings indicate that in UR and AChE the open conformation plays a key role in facilitating the entrance of the substrate to the active site, while the closed

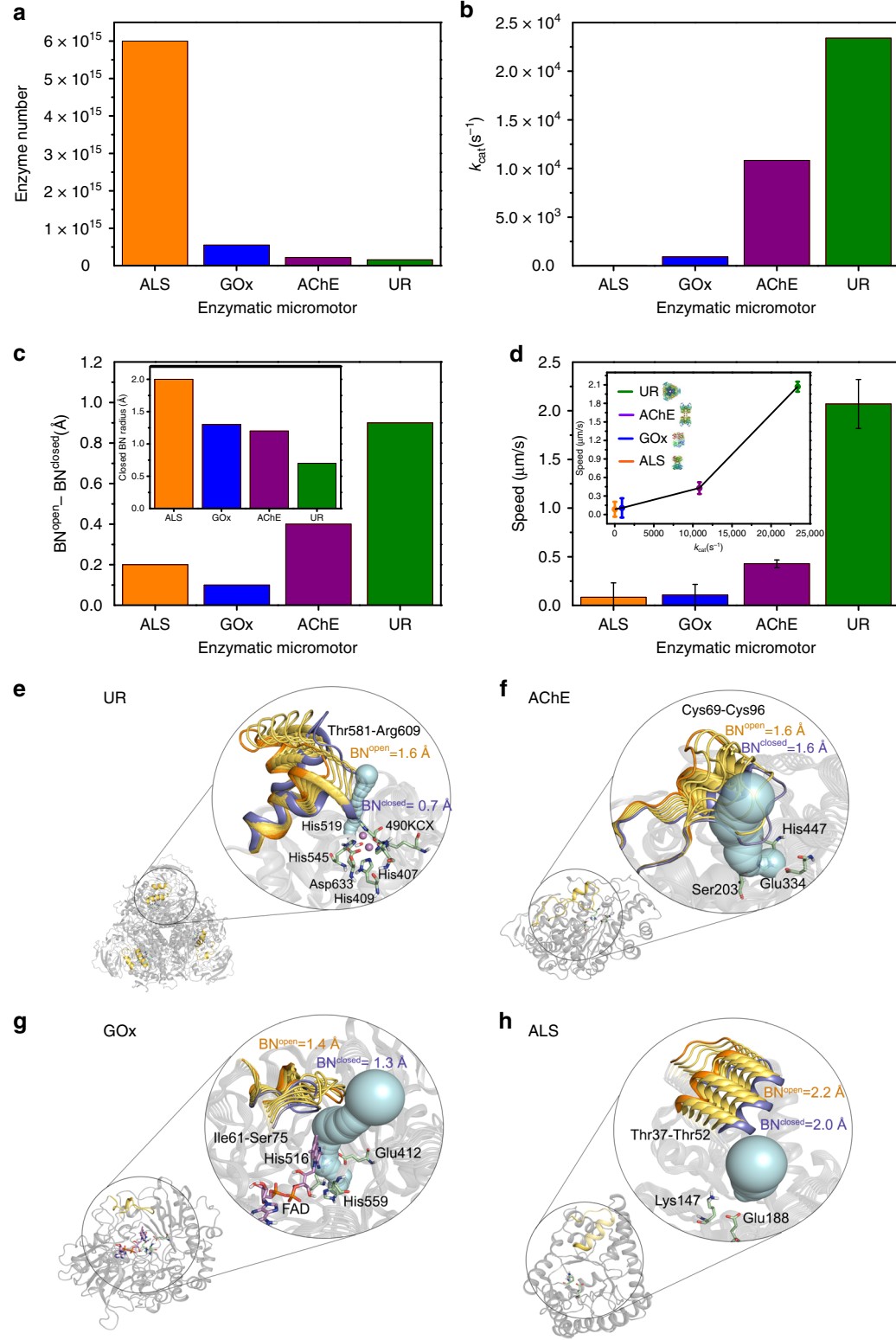

conformation tends to stabilize the substrate inside the active site for efficient catalysis. These conformational changes occurring in UR and AChE are essential over the entire course of the catalytic cycle. In contrast, the open-to-closed conformational changes in loops more distal to the active site of ALS and GOx have no significant effect on the active site tunnel and its bottleneck radius (Fig. 3e, f), indicating that these conformational changes have no

direct contribution to substrate binding and thus to catalysis. The fact that AChE and UR substrates are the smallest ones and their flexibility near the active site is higher than the rest of studied enzymes suggests that the conformational changes are required for facilitating substrate binding and product release, which in turn may influence the self-propulsion. By comparing structural and catalytic properties, we find that both the flexibility close to

**Fig. 3** Motion dynamics of HSMM as a function of enzymatic intrinsic properties. From (**a**) to (**d**) orange is used for ALS, blue for GOx, purple for AChE, and green for UR. **a** Number of enzymes attached after micromotor functionalization. **b** Literature values of the turnover number ($k_{cat}$). **c** Conformational change (in Å) of bottleneck (BN) to access the active site of each enzyme from open to closed conformation (BN$^{open}$–BN$^{closed}$) obtained through MD simulations. Inset: BN radius of each enzyme in closed conformation. **d** Average speeds of the different enzymatic HSMM for substrate concentrations that yield to maximum self-propulsion. Results are shown as the mean ± s.e.m. Twenty particles were analyzed per condition. Inset: correlation of speed and $k_{cat}$ of each enzyme. Source data are provided as a Source Data file. Enzyme structures are extracted from RCSB PDB (see Supplementary Note 3). **e**–**h** Representation of the most relevant conformational changes occurring in the MD simulations, identified through principal component analysis (PCA) for all studied enzymes. Different conformations adopted during the MD simulations by the most flexible loops are represented as open conformation in orange, intermediate conformations in dark yellow, closed conformations in blue, active site residues in green, cofactors in pink, and the active site tunnel at the open conformation of the loop in light blue. The BN (in Å) of the computed tunnel in the open/closed conformations (BN$^{open}$/BN$^{closed}$) is shown in orange/blue

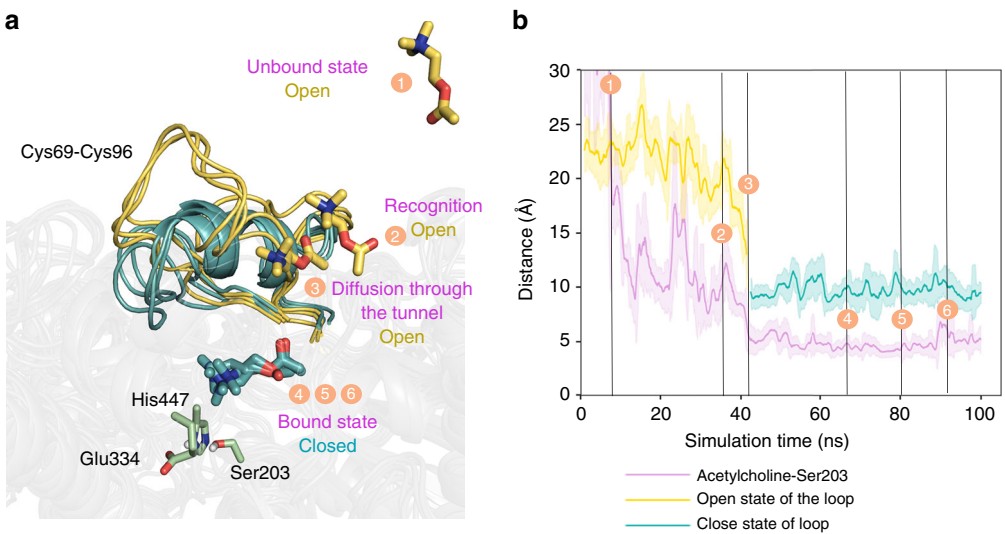

**Fig. 4** Binding mechanism of acetylcholine substrate on AChE. **a** Overlay of representative conformations along the binding pathway obtained from aMD simulations. Open conformations of the loop (defined by Cys69–Cys96 residues) and substrate conformations located outside the active site are represented in dark yellow (events 1–3), while closed conformations of the loop and substrate bound to the active site are shown in teal (events 4–6). Active site residues are shown in green. **b** Representation of the loop distance (in Å) between Pro78–Ser203 along the aMD simulation. Dark yellow solid line corresponds to open conformations of the loop, while teal color is used to highlight closed conformations. The open-to-closed transition of the loop is correlated to substrate binding, as shown by the purple solid line representing the distance (in Å) between the nitrogen atom of acetylcholine substrate and the oxygen atom of the side-chain of catalytic Ser203. Results are shown as the mean ± standard deviation (s.d.)

the active site and the turnover number positively correlate with the speed of enzymatic micromotors.

**Effect of enzyme intrinsic properties on self-propulsion.** We performed unconstrained accelerated molecular dynamics (aMD) simulations, positioning four acetylcholine substrates far from the AChE active site (at ca. 27 Å) (Fig. 4) to determine whether the flexible loop located close to the AChE active site could play a role in assisting substrate binding. The reconstructed binding pathway shows how acetylcholine enters the active site pocket through the previously identified flexible loop (binding events 1–6 in Fig. 4). When acetylcholine is located outside the active site pocket (events 1–3 in Fig. 4), the loop exhibits a high flexibility and remains in an open conformation that permits substrate binding (open conformations represented in dark yellow in Fig. 4). Once the substrate enters the active site (events 4–6 in Fig. 4), the flexibility of the loop is substantially reduced, and the loop adopts a closed conformation. This is important for productive binding of acetylcholine and to allow the hydrolysis reaction (closed conformations represented in teal in Fig. 4). These simulations indicate that prior to the chemical step, AChE undergoes open-to-closed conformational changes that are essential for substrate binding and efficient catalysis.

A similar conformational change was required for UR catalysis. In the *apo* state (i.e., enzyme in the substrate unbound state), the flap containing His593 has been reported as a likely candidate to be involved in catalysis[45]. It can adopt open conformations (flap distances of ca. 25 Å, Supplementary Fig. 16) relevant for urea binding and closed conformations (flap distances of ca. 16 Å) essential for catalysis[46,47]. This large-amplitude open-to-closed transition of the flap (*ca.* 10 Å) is required in each catalytic cycle. Such conformational changes occurring in each turnover can influence the self-propulsion of the micromotors. UR MD simulations and UR-HSMM were exposed to AHA, a reversible competitive inhibitor, to further elucidate the role of these conformational dynamics on active motion[48,49]. Compared to the *apo* state, MD simulations results indicate substantial differences in the flap conformational dynamics in the presence of AHA (Fig. 5a–c). When the inhibitor interacts with the active site of the enzyme, the wide-open conformational states of the flap are stabilized, thereby blocking the exploration of the catalytically relevant closed-states (Fig. 5b and Supplementary Fig. 17). The binding of AHA in the active site thus prevents any open-to-closed conformational transition during UR catalytic cycle (Fig. 5c). This is in agreement with the different UR X-ray structures reported in the presence of AHA (PDB 4UBP and

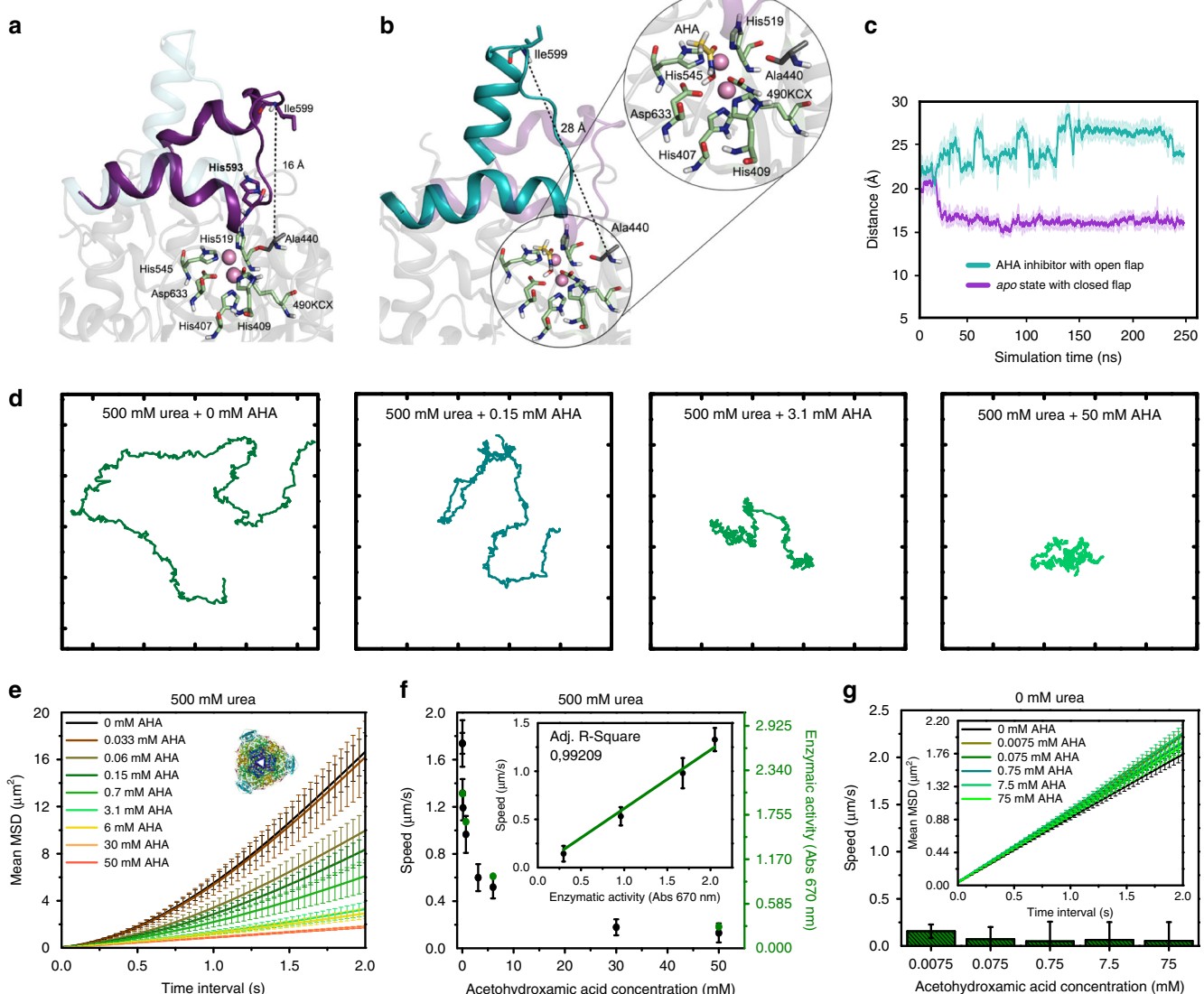

**Fig. 5** Conformation of UR and motion behavior of UR-HSMM exposed to AHA. Representative snapshots from MD simulations of **a** UR in *apo* state where the flap covering the active site can adopt a closed conformation (purple) and **b** AHA, which stabilizes wide-open conformations of the flap (teal). The zoom of the active site residues shows catalytic residues in green, nickel atoms in pink, and the AHA inhibitor in yellow. **c** MD simulated flap distance between Ala440–Ile599 in the *apo* state (purple) and AHA-bound state (teal). Results are shown as the mean ± standard deviation (s.d.). **d** Representative 28-s trajectories of UR-HSMM exposed to 500 mM urea and different concentrations of AHA (axis divided into 5 μm fragments). **e** Average MSD of UR-HSMM exposed to AHA with urea present in excess (500 mM). Enzyme structures are extracted from RCSB PDB (see Supplementary Note 3). **f** Average speed of UR-HSMM, extracted from the MSD analysis, and enzymatic activity for different AHA concentrations with urea present in excess (500 mM). Inset: correlation between speed of UR-HSMM and its enzymatic activity depending on inhibition. **g** Average speeds of UR-HSMM for different concentrations of AHA. Inset: average MSD of UR-HSMM exposed to AHA. Results are shown as the mean ± s.e.m. Twenty particles were analyzed per condition. Source data are provided as a Source Data file

1E9Y), where the flap is crystallized in an open conformation[50,51]. The knock-out of the open-to-closed transition directly impacts catalysis, which hampers the micromotor self-propulsion.

UR micromotors were exposed to increasing concentrations of AHA with urea present in excess (500 mM). The area covered by the trajectory of the micromotor decreased significantly for higher AHA concentrations (Fig. 5d, e, and Supplementary Movie 7). The increasing interaction between the inhibitor and the active site hindered urea catalysis. This was measured by analyzing the enzymatic activity of UR-HSMM exposed to different AHA concentrations using the Berthelot method (green *Y* axis in Fig. 5f and Supplementary Note 7)[52]. The speed also decreased

exponentially (Fig. 5f) and dropped by over 50% when adding 6 mM AHA, and by more than 92% when adding 50 mM AHA. Speed was positively correlated to activity (adj. *R*-square = 0.992) (inset in Fig. 5f), thus AHA exposure modified the turnover number and the results suggest a direct correlation between the catalysis rate and micromotor's speed. This agrees with the observed speed increase at higher substrate concentrations, and the speed decrease when substrate concentration is increased beyond a certain threshold (Fig. 2 and Supplementary Figs. 9 and 10). This supports the hypothesis that catalysis plays a major role in the generation of active motion and changes in the conversion rate directly affect self-propulsion.

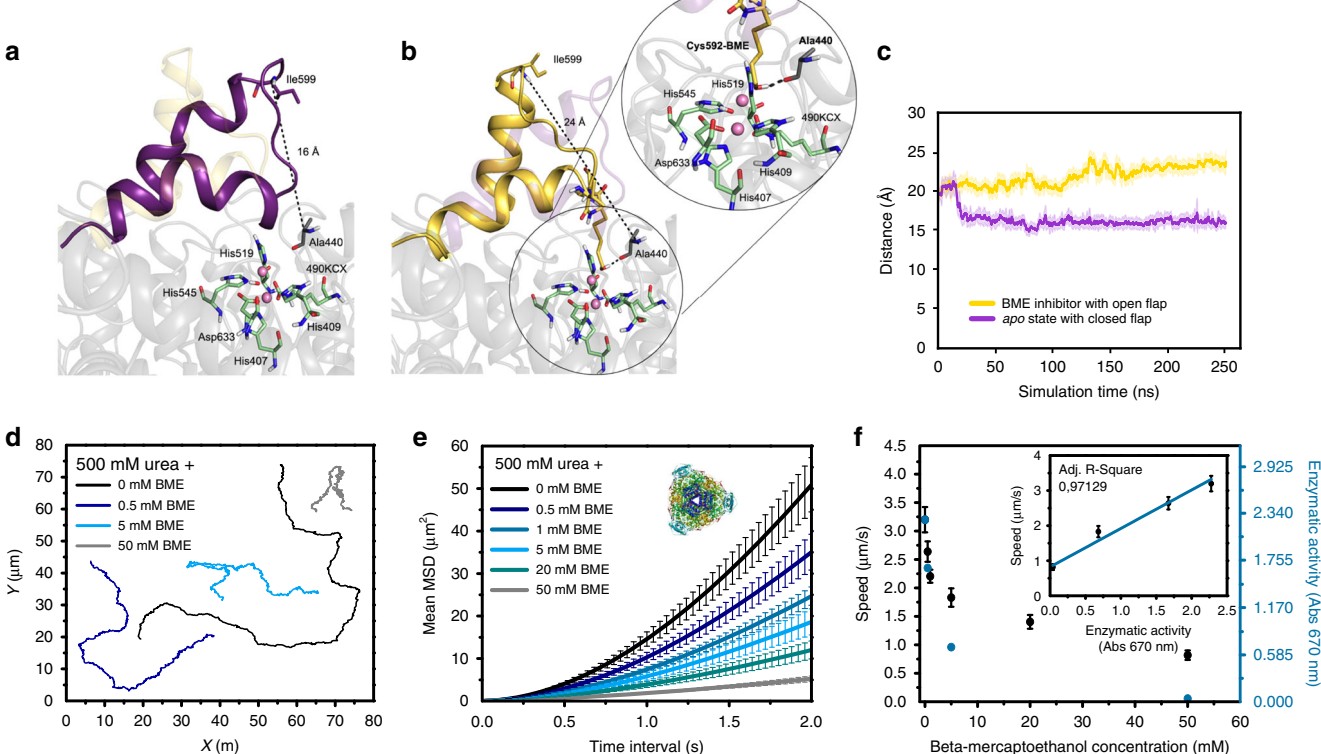

**Fig. 6** Conformation of UR and motion behavior of UR-HSMM exposed to BME. Representative snapshots taken from the MD simulations of **a** UR in *apo* state where the flap covering the active site can adopt a closed conformation (purple) and **b** BME which stabilizes more open conformations of the flap (yellow). The zoom of the active site residues shows catalytic residues in green, nickel atoms in pink, and the Cys592-BME inhibitor in yellow. **c** MD simulated flap distance between Ala440–Ile599 in the *apo* state (purple) and Cys592-BME state (yellow). Closed conformations have distances of about 16 Å while open conformations have 25 Å. Results are shown as the mean ± standard deviation (s.d.). **d** Representative trajectories of UR-HSMM exposed to 500 mM urea and different concentrations of BME. **e** Average MSD representation of UR-HSMM exposed to BME with urea present in excess (500 mM). Enzyme structures are extracted from RCSB PDB (see Supplementary Note 3). **f** Average speed of UR-HSMM, extracted from the MSD analysis and enzymatic activity for different BME concentrations with urea present in excess (500 mM). Inset: correlation of speed of UR-HSMM and its enzymatic activity depending on inhibition. Results are shown as the mean ± s.e.m. 22 Particles were analyzed for 0 and 50 mM BME, 19 particles were analyzed for 0.5 and 20 mM BME, and 21 particles were analyzed for 1 and 5 mM BME. Source data are provided as a Source Data file

Recently, the binding and unbinding of a molecule to the active site without catalysis ($k_{on}$ and $k_{off}$ in Supplementary Equations 1 and 2 in Supplementary Note 2) was reported to produce enhanced diffusion of single enzymes[10]. We examined whether these binding–unbinding interactions can produce self-propulsion of micromotors by exposing UR-HSMM to a range of AHA concentrations. No self-propulsion was detected from such interaction (Fig. 5g and Supplementary Movie 8) and any significant contribution of the binding–unbinding events on self-propulsion can be ruled out.

MD simulations showed that the effect of AHA on enzyme flexibility was crucial for inhibiting UR since the conformational change of the active site flap is a precondition for catalysis. To confirm this experimentally, the rigidity of the active site flap of UR was increased using BME. BME is known to decrease the flexibility of the UR active site flap forming a mixed di-sulfide bond with Cys592[53]. To further explore the impact of BME on the conformational dynamics of UR, we also performed MD simulations with the Cys592-BME (Fig. 6a, b and Supplementary Fig. 18). Our simulations show that BME inhibits enzymatic activity by blocking conformational changes of the flap due to H-bond interaction between Cys592-BME and the backbone of Ala440 (Fig. 6c and Supplementary Fig. 19), which also partially occupies the active site of UR (Fig. 6b). By increasing the BME concentration with urea present in excess (500 mM), the area

explored by UR micromotors diminished (Fig. 6d, e and Supplementary Movie 9) and the speed decreased exponentially (Fig. 6f). This decrease was linearly correlated (adj. $R$-square = 0.971) with the enzymatic activity obtained from the Berthelot method (inset in Fig. 6f and Supplementary Note 7)[52]. Hence, this interaction increases the rigidity of the active site flap and hampers its conformational dynamics, affecting catalysis and decreasing self-propulsion.

## Discussion

Enzyme-powered micro- and nanomotors are emerging as a very promising tool for biomedical applications[14,20,26]. Several studies have already proven their potential through the development of proof-of-concept studies on their use as cargo transport and delivery[14,34] or sensing[31,36]. However, fundamental aspects of the parameters affecting enzyme-powered motion behavior remain rather unexplored and are crucial for efficient development. Despite the fact that different studies have focused on the mechanism underlying active motion of single enzymes[3–13], it is not completely understood whether the same mechanisms are responsible for the motion of enzyme-propelled nano- and micron-sized particles. In this regard, self-diffusiophoretic mechanism has been proposed[28] but not experimentally proved. In addition, while the role of conformational changes on the enhanced diffusion of single enzymes has been proposed as a

possible mechanism[3,7–10], to the best of our knowledge, it has not been taken into account for enzymatic micro- and nanomotors.

In this manuscript, we tackle these issues studying how different intrinsic enzymatic properties affect the self-propulsion of micromotors. Initially, we showed that micromotors functionalized with enzymes with higher $k_{cat}$ displayed higher self-propelling capabilities compared to enzymes with lower $k_{cat}$. The link between the catalytic rate and active motion was also patent when increasing the substrate concentration for UR and AChE.

Moreover, we performed MD simulations to evaluate the conformational dynamics where UR and AChE (the enzymes with higher catalytic rates) displayed a higher degree of flexibility than GOx and ALS in the vicinity of the active site. MD simulations in the presence of UR and AChE substrate determined that the flexibility of specific loops located close to the active site could be crucial for assisting substrate binding, pointing out the relevance of conformational dynamics in the catalytic process.

To better understand the relationship between conformational changes, catalysis, and self-propulsion, urease micromotors were selected, since they displayed the highest motion capabilities. First, we evaluated whether catalysis plays a role on the micromotors self-propulsion by using an inhibitor (AHA) that competes with urea, binding and unbinding to the active site without being catalyzed. When both urea and AHA were present, the enzymatic activity was directly correlated to the speed. However, the lone binding and unbinding processes of AHA were not sufficient to generate micromotors' self-propulsion. Interestingly, MD simulations revealed that, apart from competing with urea, AHA increased the rigidity of the loop near the active site. To further understand how the loop conformational flexibility affects catalysis and active motion, we used BME to increase the rigidity near the active site. BME reduced the catalytic rate of urease micromotors, which was correlated with a lower speed, indicating that enzyme flexibility influenced the motion capabilities of urease micromotors.

From the aforementioned experiments we conclude that the conformational dynamics near the active site are required for urease and acetylcholinesterase catalysis, and that the rate of catalysis is essential and directly related to active motion. However, it is worth mentioning that this indirect connection between enzyme conformational dynamics and active motion at the microscale does not rule out any proposed mechanism but contributes to a better understanding of the complexity and entanglement of these intrinsic enzymatic properties and the net of causality that connects them.

Taken together, these results pave the way towards the comprehension of the processes underlying the self-propulsion of enzyme-powered micromotors. In principle, the selection of faster catalysts would lead to the fastest active motion. Although it is not clear the direct role of conformational changes on the mechanism underlying active motion, they should be always considered, and environmental conditions should be adjusted to guarantee an optimal flexibility and catalytic performance.

To further understand the mechanism behind enzyme-powered micro- and nanomotors, a larger library of enzymes should be considered to determine other physicochemical aspects that govern active motion and provide alternatives to navigate in different environments. Genetically modified enzymes with different $k_{cat}$ and structural rigidity could be tested in order to study the role of the turnover number and enzyme flexibility in more detail. Additionally, more attention should be drawn when comparing motors of different sizes, since different mechanisms may govern the motion of single enzyme motors, nano-sized or micron-sized motors.

## Methods

**Synthesis of hollow silica microcapsules**. The HSMCs were synthesized by mixing 250 μl of 2 μm particles based on PS (Sigma-Aldrich cat. no. 78452), 0.5 ml ethanol 99% (Panreac Applichem cat. no. 131086-1214), and 0.4 ml ultrapure water. Next, 25 μl ammonium hydroxide solution (Sigma-Aldrich cat. no. 221228) was added and the mixture was let to magnetically stir for 5 min. Then, 2.5 μl 3-aminopropyltriethoxysilane (APTES) 99% (Sigma-Aldrich cat. no. 440140) was added, and the reaction was let to proceed for 6 h. After, 7.5 μl tetra-ethylorthosilicate (TEOS) ≥99% (Sigma-Aldrich cat. no. 86578) was added to the solution then the reaction was let to proceed overnight. Next, the PS beads coated with a silica shell were washed with ethanol 3 times (centrifugation is always done for 3.5 min at 1503 rcf). The PS was then removed with 4 washes of dimethyl-formamide (DMF) ≥99.8% (Acros Organics cat. no. 423640010), mixing each of them for 15 min. Afterwards, the HSMCs obtained were washed 3 more times with ethanol 99% and stored.

**Functionalization of silica microcapsules with enzymes**. To synthesize hollow silica micromotors, the HSMC were washed 3 times with ultrapure water and 1 time with 1× phosphate-buffered saline (PBS) (pH = 7.4) (Thermo Fischer Scientific cat. no. 70011-036). Then, the particles were suspended in 1× PBS containing GA (2.5 wt%) (Sigma-Aldrich cat. no. G6257) and kept mixing for 3 h. Next, the HSMC functionalized with GA were washed 3 times with 1× PBS (pH = 7.4) and resuspended again in 1× PBS (pH = 7.4) with 3 mg ml$^{-1}$ of powder of urease from *Canavalia ensiformis* (Jack bean) (Sigma-Aldrich cat. no. U4002), glucose oxidase from *Aspergillus niger* (Sigma-Aldrich cat. no. G2133) or aldolase from *Oryctolagus cuniculus* (Rabbit) (Sigma-Aldrich cat. no. A2714). For acet-ylcholinesterase from *Electrophorus electricus* (Electric eel) (Sigma-Aldrich cat. no. C2888), the enzyme powder concentration used was 1 mg ml$^{-1}$. The solution was left overnight and then washed 3 times with 1× PBS (pH = 7.4). The supernatants discarded in this process were used for the total protein quantification (Supplementary Fig. 4). Then, the solution of HSMM in 1× PBS (pH = 7.4) was divided in aliquots and stored at −20 °C for further experiment.

**Fluorescent Krypton™ protein staining**. The presence of enzyme attached to the HSMC was confirmed through fluorescent Krypton™ protein staining (Thermo Scientific cat. no. 46628) (Supplementary Note 5). The enzymatic HSMM were mixed with Krypton™ protein stain solution (Thermo Scientific cat. no. 46628) diluted 10 times with ultrapure water. The mixture was shaken for 20 min to be washed once with ultrapure water after centrifugation. To observe the particles using fluorescence microscope, they were resuspended in ultrapure water. For this fluorescent dye, the wavelength of excitation is 520 nm and the detection wavelength was 580 nm. Dark conditions were needed during the entire preparation of the sample.

**Protein quantification before and after functionalization**. The presence of enzyme attached on the surface of the HSMC was confirmed indirectly through a total protein quantification of the supernatants eliminated (SN1, SN2, and SN3 in Supplementary Fig. 4) on the 1× PBS washes of the functionalization process. To perform the total protein quantification the protocols "Preparation of Standards and Working Reagents and Microplate Procedure (Sample to WR ratio = 1:8)" were followed (Supplementary Note 6) as specified in the document of Instructions of the Pierce™ BCA Protein Assay Kit (Thermo Fisher cat. no. 23227).

**Activity of urease hollow silica micromotors**. The activity in ultrapure water of urease attached to HSMC was evaluated using the Urease Activity Assay Kit (Sigma-Aldrich cat. no. MAK120) based on the Berthelot method[52]. The process followed is detailed in the "Technical Bulletin of the Urease Activity Assay Kit". It works through the ammonia generated (Supplementary Equation 3 in Supplementary Note 7) by the catalysis of urea (Sigma-Aldrich cat. no. U5128) by urease and monitoring the absorbance at 670 nm. The enzymatic activity was investigated over time, by incubating the UR-HSMM with urea for 3.5 min and analyzing the signal every 30 s through plate reader UV-spectrophotometry, using the urea solution provided in the kit dissolved in ultrapure water (Supplementary Fig. 5). The same protocol was followed to study the urease enzymatic activity exposed to different concentrations of AHA (0, 0.06, 0.7, 6, and 50 mM) (Sigma-Aldrich cat. no. 159034) and different concentrations of BME (0, 0.5, 5, and 50 mM) (Sigma-Aldrich cat. no. M6250), respectively, with urea (Sigma-Aldrich cat. no. U5128) present in excess (500 mM) all dissolved in ultrapure water, for 3 min of reaction.

**Optical video recording**. The videos of the enzymatic micromotors motion were recorded using the camera (Hamamatsu Digital Camera C11440) of an inverted optical microscope (Leica DMi8). The 63× water immersion objective was used to record the micromotors placed on a glass slide, thoroughly mixed with the water solutions of substrate at a specific concentration selected to cover the range at which these enzymes were active and showed the Michaelis-Menten growth kinetics, as reported in BRENDA, the Comprehensive Enzyme Information System (https://www.brenda-enzymes.org/). The glass slide was covered with a coverslip and videos of 25 FPS and 30–35 s were recorded up to the first 3 min after mixing.

For each enzymatic micromotor, 19–22 HSMM were recorded for each different concentration of substrate, inhibitor, and for no compound present.

**Data analysis of motion**. The videos were analyzed using a custom-designed tracking Python software to obtain the trajectories of the motion. From these, the MSD was calculated using the following:

$$\text{MSD}(\Delta t)\left\langle \sum_{i=0}^{n}(x_i(t+\Delta t)-x_i(t))^2 \right\rangle, \qquad (1)$$

where $t$ is the time and $i = 2$, for 2D analysis. The speed ($v$) was then extracted from fitting the MSD to

$$\text{MSD}(t) = 4D_t t + v^2 t^2, \qquad (2)$$

where $D_t$ is the diffusion coefficient and $v$ is the speed, as it is intended for the propulsive regime, when $t \ll \tau_r$, being $\tau_r$ the rotational diffusion time, and $t$ the time of MSD represented[39,40]. The theoretical $\tau_r$ was calculated to be $5.579 \pm 0.018$ s, which value can be understood as

$$\tau_r = \frac{1}{D_r}, \qquad (3)$$

where $D_r$ is the rotational diffusion coefficient ($D_r = 0.1792 \pm 0.0006\ \text{s}^{-1}$), which depends on the radius of the particle as it can be observed in the Stokes–Einstein equation,

$$D_r = \frac{k_B T}{8\pi\eta r^3}, \qquad (4)$$

where $k_B$ is the Boltzmann constant, $T$ is the absolute temperature, $\eta$ is the solvent viscosity, and $r$ is the radius of the diffusing particle. Hence, by extension, $\tau_r$ depends on the temperature ($T = 24 \pm 1\ °\text{C}$), the solvent viscosity ($n = 0.9107 \cdot 10^{-3}$ kg m$^{-1}$ s$^{-1}$) and the radius of the particle ($r = 1.00 \pm 0.05\ \mu\text{m}$).

**Molecular dynamics simulations**. MD simulations were used to study conformational dynamics of aldolase (ALS, PDB 1ADO), glucose oxidase (GOx, PDB 1CF3), acetylcholinesterase (AChE, PDB 1C2B), and urease (UR, PDB 3LA4). ALS, GOx, and AChE were simulated as a monomer while urease has been treated as a trimer in the MD simulations. All systems have been modeled using standard protocols described below. GOx contains a flavin adenine dinucleotide (FAD) cofactor and urease presents two nickel metal ions in the active site that required special treatment as described below. For all systems, amino acid protonation states were predicted using the H++ server (http://biophysics.cs.vt.edu/H++). In urease, two protonation states for His593 have been considered: i.e., delta protonation and doubly protonated. In the *apo* state, the flap that contains His593 reported to be likely involved in catalysis[45]. Only using the doubly protonated His593 we are able to explore the open–closed transition. The FAD cofactor has been modeled using the parameters extracted from Medvedev et al.[54]. In the case of urease in the *apo* state, the nickel atoms were treated using a non-bonded model based on ion-oxygen distance (IOD)[55]. The same non-bonded parameters are used to simulate urease with a modified cysteine residue with BME. BME is linked to Cys592 by a disulfide bridge. The parameters of BME are obtained following the AMBER tutorial for non-standard residues (http://ambermd.org/tutorials/basic/tutorial5/index.htm, see Supplementary Fig. 20 and Supplementary Table 1). To correctly model the active site of urease with the AHA covalently bound to both nickel atoms, we used the bonded model to treat the nickel atoms and the first coordination sphere on the active site of UR using the MPCB.py python program[56] of ambertools. The parameters and charges were obtained at B3LYP/6-31G* level of theory (Supplementary Fig. 21 for atom names and Supplementary Data 1 for the complete list of parameters). We also obtained parameters for the acetylcholine substrate (Supplementary Fig. 22 and Supplementary Table 2) for AChE for the MD simulations were generated within the ANTECHAMBER module of AMBER 16[57] using the general AMBER force field (GAFF)[58], with partial charges set to fit the electrostatic potential generated at the HF/6-31G(d) level by the RESP model[59]. The charges were calculated according to the Merz–Singh–Kollman scheme[60] using Gaussian 09[61].

For each system, four replicas of 500 ns of MD simulations have been carried out, while in the case of UR, 2 replicas of 250 in *apo* were performed due to the size of the system. To identify the most relevant conformational changes occurring in each enzyme, we performed MD simulations and applied the dimensionality reduction technique PCA. Dual-boost aMD simulations[62,63] were performed with four acetylcholine substrates around the AChE enzyme starting from the open conformation. Several replicas were done with a boost potential applied to all dihedrals of the system with an energy threshold of 8699.08 kcal/mol and an alpha parameter value of 380.10 while a boost potential corresponding to an energy threshold of −179,563.24 kcal/mol and alpha parameter of 10,125.76 were applied to all atoms of the system. The different systems were solvated in a pre-equilibrated truncated cuboid box with a 10 Å buffer of TIP3P water molecules using the AMBER 16 leap module. The system was neutralized by the addition of explicit counterions (Na$^+$ and Cl$^-$). All calculations were done using the ff14SB Amber force field[64]. First, the initial structures were minimized following a two-stage procedure. In the first minimization step, only solvent molecules and ions are allowed to move by restraining the positions of all atoms of both enzyme and substrates with a harmonic potential a force constant of 500 kcal mol$^{-1}$ Å$^{-2}$. In the

second step, all the atoms in the simulation cell are minimized without positional restraints. Then, the systems are heated under constant volume and periodic boundary conditions in six steps of 50 ps from 0 to 300 K in steps of 50 K. Bonds involving hydrogen atoms, including water molecules, were constrained using the SHAKE algorithm. Particle-mesh Ewald method[65] was used to account for long-range electrostatics employing an 8 Å cutoff for the treatment of Lennard-Jones and electrostatic interactions. Harmonic restraints were reduced step by step from 210 to 10 kcal mol$^{-1}$ Å$^{-2}$ along the heating process. To control the temperature a Langevin equilibration scheme was employed. Each system was then equilibrated at constant pressure of 1 atm and 300 K temperature for 2 ns using NPT ensemble. All simulations were performed with a 2 fs time step. After the equilibration process, 4 independent replicas of 500 ns for ALS, GOx, AChE and 2 independent 250 ns for UR MD were performed under the NVT ensemble and periodic boundary conditions.

Substrate access tunnels were analyzed using the standalone version of CAVER 3.0 software[66]. We selected the open and closed conformations from the PCA analysis obtained from MD simulations of all systems in the *apo* state. The starting point for the channel calculations was the active site residues. Tunnels were identified using a probe radius of 0.9 Å for ALS, GOx, and AChE while a probe radius of 0.7 Å was used for urease. Redundant tunnels were automatically removed from each structure, and the remaining ones were clustered using a threshold of 12 Å.

**Reporting summary**. Further information on research design is available in the Nature Research Reporting Summary linked to this article.

## Data availability
The authors declare that all the raw data underlying Figs. 2, 3a–d, 5d–g, 6d–f as well as Supplementary Figs. 4–14 are provided as a Source Data file and available from the corresponding author upon reasonable request.

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

## Acknowledgements

The research leading to these results has received funding from the Spanish MINECO for grants CTQ2015-68879-R (MICRODIA) and CTQ2015-72471-EXP (Enzwim). T.P. thanks MINECO for the Juan de la Cierva fellowship (FJCI-2015-25578). A.R.-R. thanks the Generalitat de Catalunya for PhD fellowship (2015-FI-B-00165). F.F. thanks the European Community for MSCA-IF-2014-EF-661160-MetAccembly grant. S.O. thanks funding from the European Research Council (ERC) under the European Union's Horizon 2020 research and innovation programme (ERC-2015-StG-679001). F.F., A.R.-R., and S.O. thank the Generalitat de Catalunya for grup emergent 2017 SGR-1707. S.S. acknowledges Foundation BBVA for the MEDIROBOTS project and the CERCA program by the Generalitat de Catalunya. The authors thank Albert Miguel-López for the development of a custom-designed Python software to track the motion of HSMC in the recorded videos. The authors acknowledge the staff at the Stuttgart Center for Electron Microscopy (StEM) and the support of Mr. Kersten Hahn with the TEM and EELS investigations. The authors are grateful for the computer resources, technical expertise, and assistance provided by the Barcelona Supercomputing Center – Centro Nacional de Supercomputación.

## Author contributions

X.A., T.P., and S.S. conceived the idea and designed the experiments. X.A. performed the experiments and analyzed the data. A.R.-R., F.F., and S.O. performed the molecular dynamics simulations and analyzed the data. All authors contributed to the discussion of results and writing the manuscript.

## Additional information

**Competing interests:** The authors declare no competing interests.

