## [Peer Review File · Nature Communications]

Reviewers' comments:

Reviewer #1 (Remarks to the Author):

The authors present experimental and simulation data on how the type of enzyme is affecting the observed mobility of swimmers equipped with these enzymes. The reported results provide an interesting opportunity to get access to a predictive tool to select enzymes that are likely to be efficient swimmers. It is overall a well-written paper with attractive figures.

Specific comments:

- The rationale of the chosen 4 enzymes should be outlined more clearly at an early stage of the manuscript
- The amount of remaining activity of the enzymes immobilized on the particles should be specifically mentioned in the main text i.e., what is the percentage of active enzymes on the particle surface? Is there any chance that this has an impact on their findings?
- How were the 20 particles used to determine the mobility properties selected?

Reviewer #2 (Remarks to the Author):

This paper reports very interesting experiments, augmented by computation, that explore the origins of enhanced diffusion (and in the present experimental context of hollow silica particles self-propulsion). Unfortunately the authors make overblown claims that are in no way supported by the data or results. The opening line of the conclusion section is

"In this manuscript we demonstrated that the catalytic step and coupled conformational changes are essential for the self-propulsion of micromotors, through a combination of experimental and simulation analyses."

They did no such thing, they showed that of the four enzymes studied, the two with large conformational changes had the largest "self-propulsion".

The phrase self-propulsion itself is questionable - in what direction is the particle propelled, and what is the frame of reference? Does the direction of motion (if there is any - not clear from the paper) have anything to do with the hole in the particle. Is the microscopy sufficiently well resolved to determine correlation between orientation of the particle and instantaneous direction of motion.

The authors state that GOx-HSMM and ALS-HSMM only showed Brownian motion. This suggests that Urease and Ach showed other than Brownian motion. If this is the claim, what is the statistical measure distinguishing the two motions, or is it only based on the magnitude of the MSD.

This paper will be of interest to a wide community of scientists, and will influence thinking in the field. The data and computations themselves are persuasive. I strongly recommend the authors to go through every claim made and ask whether in fact the limited data support the sweeping conclusions. The four enzymes studied here, with significant differences in both catalytic rate and conformational flexibility are sufficient to make a case in support of the importance of conformational changes in self-propulsion by immobilized enzymes, but further studies (as suggested by the authors in the conclusion section) will be necessary to establish the case more firmly. I would also really like to see more discussion on what the authors mean by self-propulsion and how it differs from enhanced Brownian motion.

R. Dean Astumian

Reviewer #3 (Remarks to the Author):

The manuscript by Arque et al concerns experimental work that is designed to shed light on enhanced diffusion of single enzymes, by attaching them to the surface of larger colloids that can be directly studied using particle tracking. The experimental work is complemented with computational work that examines the conformational changes of the enzymes used in the experiments.

I would like to start by saying that this work belongs to a very relevant and active field of research and careful contributions that examine various aspects of the problem will be very helpful in unravelling the mechanisms behind the observed activity of enzymes. The work by Arque et al is an interesting contribution in this direction. However, in my view the actual work that is done does not support the conclusions of the manuscript, and I would like to invite the authors to thoroughly re-examine their work to elevate it to a worthwhile contribution to the field.

I mention a number of major concerns below:

- They state "we demonstrated that the catalytic step and coupled conformational changes are essential for the self-propulsion of micromotors, through a combination of experimental and simulation analyses" and "Molecular dynamics simulations were crucial to establish a connection between enzymatic flexibility and active motion at the microscale". I do not think that their results are sufficient to establish these relations. Experimentally, they have observed a correlation between catalytic activity and propulsion. In simulations, they have observed that conformational changes are necessary for successful catalysis (in urease). These two results together are not sufficient to claim a direct connection between conformational changes and propulsion, i.e. "swimming". For example, these results do not rule out a self-phoretic mechanism, which would depend on successful catalysis, but works independently of conformational changes. If the authors look into Ref. 10, it is indeed predicted (proposed) that attaching acetylcholinesterase to a micron-sized bead will lead to propulsion with the order of magnitude of propulsion estimated to be of the order of the observed results. It is not clear to me why the authors do not discuss this obvious link to a theoretical proposal from 14 years ago they cite in the paper.

- They state "We also observed that the binding and unbinding processes are incapable of producing micromotor self-propulsion". However, this is referring to the observation that binding-unbinding of the inhibitor did not cause self-propulsion. They did not show whether or not binding-unbinding of the substrate (which they showed requires a conformational change as opposed to inhibitor binding which doesn't) can lead to self-propulsion. For this, they would have to use a non-competitive inhibitor, i.e. one that affects the rate k_{cat} , but does not affect binding and unbinding of the substrate.

Other comments:

- They use hollow microcapsules, whereas in other studies they (e.g. Ref 27) and others have used solid capsules. Can they comment on the difference in propulsion observed between the two, in similar conditions (urease+urea?)

- Description of the MD simulations in the SI: why are aldolase, GOx and AChE treated as monomers, while urease is treated as a trimer? In which multimeric conformation are the enzymes expected to be in the experiments and why? Could this choice affect the results for the conformational flexibility of the enzymes? Moreover, the molecular weights cited on page 9 of the main text would also depend on the choice of multimer. It seems like the authors should discuss these choices in the main text.

What are the thermodynamic properties of the enzymes? Change in temperature?

It's not obvious to me which symmetry is broken to achieve the directed motion? Do they have a patchy structure of the enzymes or a smooth uniform covering? If the latter, how was a monolayer coverage of the shell ensured?

In Fig.2(c) why is the range of concentration of GLC so different to the other substrates?

To test the effect of conformational changes, why not study the differences that arise for different substrates of one enzyme? For example, the 2 substrates of aldolase as in Rago et. al 2015. If force is proportional to k_{cat} , compare UR-HSMM with 1 UR attached to the shell with ALS-HSMM with 1800 ALS attached, GOx-HSMM with 25 GOx attached and AChE-HSMM with 2 AChE; hence, two orders of magnitude difference between ALS and UR. However, Fig. 3 and Supplementary Fig. 13 and 14 do not show two orders of magnitude difference in the number of UR and ALS the silica shell is functionalized with.

Under what conditions are the conformational changes measured in the MD simulations?

Is it possible that a higher flexibility was found in UR and AChE because they are bigger?

In reference to conformational changes, the suggestion that "In ALS and GOx, these changes are decoupled from catalysis and act unilaterally" and also the suggestion that conformational changes in ALS and GOx have "no direct contribution to substrate binding and thus catalysis" is not justified, as it is in contradiction with Rago et. al. 2015.

One final comment about giving credit: stochastic swimming has been discussed in an extensive literature, which is mentioned and expanded on in Ref. 6 in addition to discussing the collective heating mechanism that the manuscript currently mentions. It is not appropriate to give the credit about this concept to Ref. 7, as it is a review article and is discussing other people's works.

Reviewers' comments:

REVIEWER #1 (Remarks to the Author):

This reviewer commented positively about our manuscript noting that “The reported results provide an interesting opportunity to get access to a predictive tool to select enzymes that are likely to be efficient swimmers. It is overall a well-written paper with attractive figures”.

He/she suggested publication after consideration of the following issues.

Comment 1: The rationale of the chosen 4 enzymes should be outlined more clearly at an early stage of the manuscript.

Response 1: We thank the reviewer for this comment. We have added an additional sentence in the introduction (page 4, line 9) to clarify the reasons behind the selection of the 4 different enzymes and to improve the clarity and understanding of the manuscript.

“These 4 enzymes were selected due to their differences in k_{cat} , to understand whether the catalytic turnover plays an important role on the self-propulsion of micromotors.”

Comment 2: The amount of remaining activity of the enzymes immobilized on the particles should be specifically mentioned in the main text i.e., what is the percentage of active enzymes on the particle surface? Is there any chance that this has an impact on their findings?

Response 2: We thank the reviewer for pointing this out. It would be of special relevance to know the percentage of active enzymes on our micromotors surface. However, to the best of our knowledge, current techniques to measure enzymatic activity and kinetics do not allow for the direct measurement of the percentage of *active* enzymes immobilized on the particle surface. We believe that this particular concern does not have a special impact in our findings since we demonstrate that all 4 enzymes show activity after being conjugated to the particle surface (See supporting information, Figures S5-S8). In addition, several works have demonstrated that enzymes immobilized through glutaraldehyde linker onto a surface retain enzymatic activity and the improved stabilization even allows for conditions that improve enzymatic activity.¹⁻⁴

Moreover, we would like to highlight the fact that increasing the number of enzymes in the case of urease did not affect the motion behaviour. The number of enzymes was shown to be relevant in Patino et al., JACS 2018, but it is also demonstrated that the speed is stabilized when surpassing a specific threshold of enzyme number. We observed the same when increasing the quantity of urease attached, the speed was not modified. This shows that, when having enough enzymes, it is not the overall enzymatic activity of all the enzymes attached to the particle that modifies active motion but the catalytic turnover of each enzyme specifically.

1. Bornscheuer, U. T. Immobilizing enzymes: How to create more suitable biocatalysts. *Angew. Chemie - Int. Ed.* **42**, 3336–3337 (2003).

2. Mateo, C., Palomo, J. M., Fernandez-lorente, G., Guisan, J. M. & Fernandez-lafuente, R. Improvement of enzyme activity , stability and selectivity via immobilization techniques. *Enzyme Microb. Technol.* **40**, 1451–1463 (2007).
3. Barbosa, O. *et al.* Glutaraldehyde in bio-catalysts design: a useful crosslinker and a versatile tool in enzyme immobilization. *RSC Adv.* **4**, 1583–1600 (2014).
4. López-Gallego, F. *et al.* Enzyme stabilization by glutaraldehyde crosslinking of adsorbed proteins on aminated supports. *J. Biotechnol.* **119**, 70–75 (2005).

Comment 3: How were the 20 particles used to determined the mobility properties selected?

Response 3: To avoid any type of bias on the results, we randomly selected 20 particles in each condition, recorded them for 30-35s and performed optical tracking, the mean squared displacement and speed calculations through an automatic and custom-designed tracking Python software.

REVIEWER #2 (Remarks to the Author):

This reviewer highlighted that “this paper reports very interesting experiments” and that “This paper will be of interest to a wide community of scientists, and will influence thinking in the field.”. Yet, he was more critical with our manuscript, mainly regarding the conclusions section. After addressing all his comments, we believe that the manuscript improved the quality and clarity. We report below the comments of this reviewer and our responses to them.

Comment 1: The opening line of the conclusion section is "In this manuscript we demonstrated that the catalytic step and coupled conformational changes are essential for the self-propulsion of micromotors, through a combination of experimental and simulation analyses." They did no such thing, they showed that of the four enzymes studies, the two with large conformational changes had the largest "self propulsion".

Response 1: We thank the reviewer for this comment. We have now changed the Conclusions for a Discussion section for format reasons. We have made some modifications based on the reviewer comment as it can be seen in page 21, line 14:

“In this manuscript we tackle these issues studying how different intrinsic enzymatic properties affect the self-propulsion of micromotors. Initially, we showed that micromotors functionalized with enzymes with higher k_{cat} displayed higher self-propelling capabilities compared to enzymes with lower k_{cat} . The link between the catalytic rate and active motion was also patent when increasing the substrate concentration for UR and AChE.

Moreover, we performed MD simulations to evaluate the conformational dynamics where UR and AChE (the enzymes with higher catalytic rates) displayed a higher degree of flexibility than GOx and ALS in the vicinity of the active site. MD simulations in the presence of UR and AChE substrate determined that the flexibility of specific loops located close to the active site could be crucial for assisting substrate binding, pointing out the relevance of conformational dynamics in the catalytic process.”

Or in the page 22, line 16:

“From the aforementioned experiments we conclude that the conformational dynamics near the active site are required for urease and acetylcholinesterase catalysis, and that the rate of catalysis is essential and directly related to active motion. However, it is worth mentioning that this indirect connection between enzyme conformational dynamics and active motion at the microscale does not rule out any proposed mechanism but contributes to a better understanding of the complexity and entanglement of these intrinsic enzymatic properties and the net of causality that connects them.”

Comment 2: The phrase self propulsion itself is questionable - in what direction is the particle propelled, and what is the frame of reference?

Response 2: We thank the reviewer for this comment. We would like to note that the self-propulsion can be observed in the supplementary videos VS1, VS2, VS7 and VS8 included in the initial submission (and in the Videos Response 1-3 provided with the Response Letter). The dynamics show a significant difference from Brownian particles and for high substrate

concentrations (and low inhibitor concentration) a directed motion over long period (29 seconds). Since the particles have an asymmetric distribution of enzymes (thoroughly studied in Patino et al., JACS 2018), the most plausible reason for the quadratic MSD originates from the asymmetric chemical gradient of products leading to self-propulsion.

Self-propulsion is a widely used term that refers to the autonomous motion of artificial or natural objects without the need of external power sources. In this manuscript, the propulsion of micromotors is achieved by the conversion of substrates into products, mediated by enzymes. Since there are no external power sources, we refer to their movement using the concept “self-propulsion” or conjugations of the verb “self-propel”, since is a common terminology used in the field of catalytic nano- and micromotors also by other groups (See references 3, 6, 7, 10, 12, 15, 17-19, 21, 24, 31-33, 38 from the main text of the manuscript).

Regarding the second part of the question, i.e. the directionality of propelled particles, one needs to consider that at the micro- and nanoscale particles are subjected to Brownian forces, which randomize the direction of the particle. The rotational diffusion time (average time for angle randomization) depends on the particle radius (Ref 38 in main text). Larger particles result in a higher rotational diffusion time, where the motion will appear more “directional” than smaller particles for the same period of time. Said this, our particles maintain their direction of motion for about 6 seconds.

In our case, particles do not display a visible frame of reference on the surface, but a small defect can be spotted in few cases which can be used as contrast point to check the direction of the motion. By tracking the particle orientation, it is confirmed that the particles are not rotating while maintaining a straight trajectory (Videos Response 1-3, urease micromotors, 500 mM urea). It is only when the particle turns that then the direction of motion is affected in the same order.

Comment 3: Does the direction of motion (if there is any - not clear from the paper) have anything to do with the hole in the particle. Is the microscopy sufficiently well resolved to determine correlation between orientation of the particle and instantaneous direction of motion.

Response 3: To determine the directionality of the particles, a contrast reference is needed. In this case, microscopy resolution does not allow to distinguish the hole in the particle. Authors believe that determining whether the hole in the particle capsule can influence the directionality of the particles could be of important relevance in this type of micromotors and will be the topic of future research. However, current methodology and technical approaches do not allow to experimentally assess that effect. In the present manuscript, we thoroughly characterized the number of holes per particle resulting in a mean number of 1.2 ± 0.1 holes/particle, as indicated in page 5, lines 16. Also, the effect of having a hole (in case there was an effect) would influence all samples at the same level, meaning that its impact on the reported results is not relevant.

In this response we would like to refer to the additional videos provided in Response 3 (Videos Response 1-3). In a few cases, we localized a small defect on the particle surface that can be used to know the orientation of the particle, since the hole is not visible. Results show that the small defect is always maintained at the same side while swimming straight. Thus, the orientation of the particle is directly related to the particle rotating every 6-9 seconds.

Comment 4: The authors state that GOx-HSMM and ALS-HSMM only showed Brownian motion. This suggests that Urease and Ach showed other than Brownian motion. If this is the claim, what is the statistical measure distinguishing the two motions, or is it only based on the magnitude of the MSD.

Response 4: We thank the reviewer for this relevant inquiry. As mentioned in Response 2 and 3, we have shown the self-propulsion in the supplementary videos VS1 (urease motors + urea), VS2 (acetylcholinesterase motors + acetylcholine), VS7 (urease motors + urea + acetohydroxamic acid) and VS8 (urease motors + urea + β -mercaptoethanol), and in the Response Videos 1-3 (urease motors + urea), clearly different from the Brownian motion in the supplementary videos VS3 (glucose oxidase motors + glucose) and VS4 (aldolase motors + fructose 1,6-bisphosphate). Indeed, this difference is based on the magnitude of the MSD and whether it is defined by a linear or quadratic equation. Since the particles have an asymmetric distribution of enzymes (studied in Patino et al., JACS 2018), self-propulsion is the most plausible explanation for a parabolic MSD.

To extensively clarify this point, we would like to refer to a recent publication, where authors reviewed the state of the art on the fundamental aspects of enzyme-powered micro- and nanomotors (Patino et al., Accounts, 2018), and reported some hints on how to perform the analysis of particle motion and how to distinguish ballistic motion from Brownian motion, which was defined as follows:

“The dynamics of self-propelled particles are usually analyzed by calculating the mean squared displacement (MSD) of their positions over time.^{1,2} By assuming a constant speed over time and randomization of the particle’s position and orientation due to Brownian fluctuations, one can obtain the following MSD:

$$MSD(\Delta t) = \langle (\vec{r}(\Delta t) - \vec{r}(0))^2 \rangle = 4D_t \Delta t + \frac{v^2 \tau_r^2}{2} \left[\frac{2\Delta t}{\tau_r} + e^{-2\Delta t/\tau_r} - 1 \right],$$

where $\vec{r}(0)$ is the position of the particle at the initial time, $\vec{r}(\Delta t)$ the position of the particle after a time Δt , D_t the translational diffusion coefficient, τ_r the rotational diffusion time and v the speed of the particle. This well-known equation only applies when particle dynamics are characterized by constant speed and the particle experiences no torques. However, we can distinguish two different regimes that simplify this equation. At longer time scales ($\Delta t \gg \tau_r$), it can be written as:

$$MSD(\Delta t) = (4D_t + v^2 \tau_r) \Delta t = 4D_e \Delta t,$$

which is analogous to the case of a passive Brownian particle and is referred to as enhanced diffusion. At shorter time scales ($\Delta t \ll \tau_r$), it takes the form:

$$MSD(\Delta t) = 4D_t \Delta t + v^2 \Delta t^2,$$

which is called the propulsive or ballistic regime, since we should see an effective directional movement where the particle seems to continuously propel in a specific direction.

These equations are commonly used for the motion analysis of catalytic and bio-catalytic (enzymatic) micro- and nanoswimmers, since they can give statistically averaged results. The shape of the MSD curve can change depending on the size of the particle, as the rotational diffusion time increases with the cube of its size (Figure R4).¹ For nano-particles, the rotational

diffusion time is very small compared to the time resolution of typical equipment and, therefore, only the enhanced diffusion regime can be observed (Figure R4A). In these cases, one can only obtain an enhanced or effective diffusion by fitting the experimental MSD to a linear function. For near micron-sized particles, a short propulsive (quadratic) regime should theoretically be visible, although it might appear hidden if the time resolution is low (Figure R4B). For micro-particles, the rotational diffusion time lays within the observable time. In this case, the propulsive regime appears at times shorter than the rotational diffusion time and the diffusive regime at longer times (Figure R4C).

Figure R4. Dynamics of micro- and nanoswimmers. (top) MSD simulations of active Brownian particles of (A) $\varnothing = 350$ nm,³ (B) $\varnothing = 800$ nm,⁴ and (C) $\varnothing = 2$ μ m,⁵ moving at three different speeds. (bottom) Real MSD data from self-propelled particles powered by the decomposition of urea by urease. Panel A reproduced with permission from ref 12. Copyright 2018 Wiley. Panel B reproduced with permission from ref 29. Copyright 2015 American Chemical Society. Panel C reproduced with permission from ref 26. Copyright 2016 American Chemical Society. (Figure 6 from Patino et al., *Accounts*, 2018)

1. Howse, J. R. *et al.* Self-Motile Colloidal Particles: From Directed Propulsion to Random Walk. *Phys. Rev. Lett.* **99**, 048102 (2007).
2. Dunderdale, G., Ebbens, S., Fairclough, P. & Howse, J. Importance of Particle Tracking and Calculating the Mean-Squared Displacement in Distinguishing Nanopropulsion from Other Processes. *Langmuir* **28**, 10997–11006 (2012).
3. Hortelão, A. C., Patiño, T., Perez-Jiménez, A., Blanco, À. & Sánchez, S. Enzyme-Powered Nanobots Enhance Anticancer Drug Delivery. *Adv. Funct. Mater.* **28**, 1705086 (2018).
4. Dey, K. K. *et al.* Micromotors Powered by Enzyme Catalysis. *Nano Lett.* **15**, 8311–8315 (2015).
5. Ma, X., Wang, X., Hahn, K. & Sánchez, S. Motion Control of Urea-Powered Biocompatible Hollow Microcapsules. *ACS Nano* **10**, 3597–3605 (2016)."

Comment 5: This paper will be of interest to a wide community of scientists, and will influence thinking in the field. The data and computations themselves are persuasive. I strongly recommend the authors to go through every claim made and ask whether in fact the limited data support the sweeping conclusions. The four enzymes studied here, with significant differences in both catalytic rate and conformational flexibility are sufficient to make a case in support of the importance of conformational changes in self-propulsion by immobilized enzymes, but further studies (as suggested by the authors in the conclusion section) will be necessary to establish the case more firmly. I would also really like to see more discussion on what the authors mean by self-propulsion and how it differs from enhanced brownian motion.

Response 5: Authors thank the reviewer for this valuable comment. We are happy to read that the referee is very positive about the impact our simulations can make in the community and how this paper will be of interest for a broad community. We are glad to see that he believes that our data is “sufficient” to make a case. Regarding the discussion, we have modified the conclusions (now replaced by the Discussion section for format reasons) accordingly, as detailed in Response 1.

We have discussed the differences of self-propulsion to enhanced Brownian motion in a previous question. We believe that after addressing these concerns, the clarity of the manuscript has been significantly improved.

R. Dean Astumian

REVIEWER #3 (Remarks to the Author):

This reviewer commented positively on our manuscript, noting that “this work belongs to a very relevant and active field of research and careful contributions that examine various aspects of the problem will be very helpful in unravelling the mechanisms behind the observed activity of enzymes. The work by Arque et al is an interesting contribution in this direction.” Below are the answers to his/her comments:

Comment 1: They state “we demonstrated that the catalytic step and coupled conformational changes are essential for the self-propulsion of micromotors, through a combination of experimental and simulation analyses” and “Molecular dynamics simulations were crucial to establish a connection between enzymatic flexibility and active motion at the microscale”. I do not think that their results are sufficient to establish these relations. Experimentally, they have observed a correlation between catalytic activity and propulsion. In simulations, they have observed that conformational changes are necessary for successful catalysis (in urease). These two results together are not sufficient to claim a direct connection between conformational changes and propulsion, i.e. “swimming”. For example, these results do not rule out a self-phoretic mechanism, which would depend on successful catalysis, but works independently of conformational changes.

Response 1: The authors thank the reviewer for noticing this. We have modified the conclusions section (now replaced by the Discussion section for format reasons) as detailed in Response 1 to Reviewer #2. In addition, we would like to clarify that in the present manuscript, we do not claim a direct connection between conformational changes and propulsion, as Reviewer #3 mentions in this comment. We demonstrate by MD simulations that conformational changes occurring at the vicinity of the acetylcholinesterase and urease active sites are a requirement for catalysis. In acetylcholinesterase, substrate binding favors closed conformations of the active site loop for efficient catalysis. The presence of inhibitors (acetohydroxamic acid and β -mercaptoethanol) in urease decreases the flexibility of the active site flap, thus preventing its catalytic activity. Therefore, the role of conformational changes and catalysis, in this specific case, cannot be decoupled and taken separately.

As noted by the reviewer, conformational changes could be required for catalysis but not the direct causality of movement. We modified the main text of the manuscript as suggested by the reviewer to improve the clarity of these important concepts in page 22, line 16:

“From the aforementioned experiments we conclude that the conformational dynamics near the active site are required for urease and acetylcholinesterase catalysis, and that the rate of catalysis is essential and directly related to active motion. However, it is worth mentioning that this indirect connection between enzyme conformational dynamics and active motion at the microscale does not rule out any proposed mechanism but contributes to a better understanding of the complexity and entanglement of these intrinsic enzymatic properties and the net of causality that connects them.

Taken together, these results pave the way towards the comprehension of the processes underlying the self-propulsion of enzyme-powered micromotors. In principle, the selection of faster catalysts would lead to fastest active motion. Although it is not clear the direct role of conformational changes on the mechanism underlying active motion, they should be always considered, and environmental conditions should be adjusted to guarantee an optimal flexibility and catalytic performance.”

Comment 2: If the authors look into Ref. 10, it is indeed predicted (proposed) that attaching acetylcholinesterase to a micron-sized bead will lead to propulsion with the order of magnitude of propulsion estimated to be of the order of the observed results. It is not clear to me why the authors do not discuss this obvious link to a theoretical proposal from 14 years ago they cite in the paper.

Response 2: We appreciate the reviewer comment and it is indeed true that Golestanian et al. in Phys. Rev. Lett., 2005 (Ref. 10) proposed the case of propulsion of a micron-sized beads where acetylcholinesterase is attached on the surface. However, there are certain relevant aspects that differ from our system and make difficult the comparability. In the case proposed the particles are $R = 2 \mu\text{m}$ instead of $R = 1 \mu\text{m}$ of our motors, and the speed obtained is 1 nm/s instead of 500 nm/s, as our experimental data shows. Additionally, the system presented in Ref. 10 is based on having a single and localized enzymatic site, while the enzymatic distribution on the surface of the silica microcapsules we are presenting in this manuscript is non-homogeneous but covering the particles with several enzymatic patches. Unfortunately, it doesn't seem accurate to accept the comparability of results of this manuscript with the proposed situation in Golestanian et al., Phys. Rev. Lett., 2005.

Comment 3: They state "We also observed that the binding and unbinding processes are incapable of producing micromotor self-propulsion". However, this is referring to the observation that binding-unbinding of the inhibitor did not cause self-propulsion. They did not show whether or not binding-unbinding of the substrate (which they showed requires a conformational change as opposed to inhibitor binding which doesn't) can lead to self-propulsion. For this, they would have to use a non-competitive inhibitor, i.e. one that affects the rate k_{cat} , but does not affect binding and unbinding of the substrate.

Response 3: We thank the reviewer for this valuable comment. We changed the sentence "We also observed that the binding and unbinding processes are incapable of producing micromotor self-propulsion" by "However, the lone binding and unbinding processes of AHA were not sufficient to generate micromotors' self-propulsion." in page 22, line 8. Acetohydroxamic acid (AHA) directly competes with the urea substrate for the metal centre in the active site of urease as shown in the X-ray structures (PDB codes: 4UBP and 1E9Y). In contrast, β -mercaptoethanol (BME) reacts with some of the cysteine residues of the urease flap (not located on the active site pocket), which were suggested to affect its conformational dynamics (PDB: 3LA4). Motivated by these observations, we decided to investigate BME inhibition with MD simulations. We observed that BME alters the flap conformational dynamics and favours catalytically non-productive open conformations. Although BME does not compete with urea substrate for the active site of the enzyme, the hydroxyl group of the alkyl chain of BME makes a hydrogen bond with the backbone of Ala440, which is situated quite close to the catalytic centre (*ca.* 7 Å) and might block the entrance of urea in the active site. Therefore, both AHA and BME influence the flap conformational dynamics and also the active site pocket of the enzyme, although via a different inhibition mechanism. As suggested by the referee the use of an allosteric inhibitor could provide relevant new insights on the effect of (un)binding of the urea substrate. However, as we were not able to find any allosteric inhibitor in the literature, we decided to test our hypothesis using BME that uses a different mechanism compared to AHA.

Other comments:

Comment 4: They use hollow microcapsules, whereas in other studies they (e.g. Ref 27) and others have used solid capsules. Can they comment on the difference in propulsion observed between the two, in similar conditions (urease+urea?)

Response 4: Regarding the solid capsules, in Patino et al., JACS 2018 we reported a ballistic motion with average speed of 6 $\mu\text{m/s}$. We have also reported the use of the same hollow capsules either non-Janus (Patino et al., 2019 Nanoletters, Mean Speed = 6 $\mu\text{m/s}$) or Janus (Ma et al 2016 ACS Nano, Mean Speed = 8 $\mu\text{m/s}$). In the present work, we describe an average speed of 2-3 $\mu\text{m/s}$. Therefore, in all cases, the same ballistic behaviour and similar speed (same order of magnitude) have been observed.

Comment 5: Description of the MD simulations in the SI: why are aldolase, GOx and AChE treated as monomers, while urease is treated as a trimer? In which multimeric conformation are the enzymes expected to be in the experiments and why? Could this choice affect the results for the conformational flexibility of the enzymes?

Response 5: We thank the reviewer for the comment. Aldolase is known to be active as both monomer and dimer experimentally (see: *Acta Cryst.* **2008**, D64, 543), GOx is also usually modelled as monomer as the formation of dimer is promoted after glycosylation (*ACS Catal.* **2017**, 7, 6188), and AChE has been modelled both as dimer and monomer (*J. Biomol. Struct. Dyn.* **2010**, 28, 393). As shown in the Figure R5 below, ALS, GOx, and AChE present relevant conformational changes taking place in regions not directly involved in the protein-protein interface, therefore, to reduce the computational cost associated to these simulations we decided to focus on the monomeric form of all these enzymes. We do not expect a significant change on enzyme flexibility specially in the active site region located far away from the interface. Urease is a more complex enzyme that presents an oligomerization state of dimer of trimers (PDB: 3LA4). In this case, as the monomers forming the trimer are intertwined the simulation of the monomer is not appropriate as the structure and shape of the enzyme would not be preserved. As done in previous computational studies, we simulated the trimeric state of urease (*J. Am. Chem. Soc.* **2012**, 134, 6). It should be also noted that we have used previously described protocols to perform the MD simulations of urease (*J. Am. Chem. Soc.* **2012**, 134, 6).

a. Aldolase (ALS)

b. Glucose Oxidase (GOx)

c. Acetylcholinesterase (AChE)

Figure R5. Representation of the most relevant conformational changes observed along the MD trajectories for the monomeric states (shown in yellow). The oligomeric X-ray state is represented in blue (PDB codes: 1ADO, 4UDQ, 5DTI for ALS, GOx, and AChE, respectively), and overlaid to the simulated monomer (shown in grey). As can be shown in the figure, the most relevant conformational changes (in yellow) are located far away from the protein-protein interfaces. The X-ray structure used to overlay GOx on the dimeric state corresponds to a 5-hydroxymethylfurfural oxidase as there are no dimeric structures available for GOx (PDB code: 1CF3).

Comment 6: Moreover, the molecular weights cited on page 9 of the main text would also depend on the choice of multimer. It seems like the authors should discuss these choices in the main text.

Response 6: The molecular weights chosen were extracted from the datasheet of the company from which it was bought: Urease from *Canavalia ensiformis* (Jack bean) (Sigma-Aldrich cat. no. U4002) is a hexamer of 430-480 kDa as a major protein form, glucose oxidase from *Aspergillus niger* (Sigma-Aldrich cat. no. G2133) is a dimer of 160 kDa and acetylcholinesterase from *Electrophorus electricus* (Electric eel) (Sigma-Aldrich cat. no. C2888) is a tetramer of 230-260 kDa. Aldolase from *Oryctolagus cuniculus* (Rabbit) (Sigma-Aldrich cat. no. A2714) did not have a datasheet available but it is known to be a homotetramer of 150 kDa (Marsh and Lebherz 1992).

As suggested by the reviewer, we have added this information in the main text by modifying the sentence by “size and weight (provided by the purchasing company and extracted from literature) resulted in the highest micromotor’s speed, following this increasing order: ALS (150 kDa) and GOx (160 kDa), AChE (230-280 kDa), and UR (440-480 kDa)” (page 9, line 22).

Comment 7: What are the thermodynamic properties of the enzymes? Change in temperature?

Response 7: This point is of great interest in order to understand the relevance of reaction enthalpy for the self-propulsion of enzymatic micromotors. In the present manuscript we did not study this aspect since our focus was on the catalytic turnover and the conformational changes. However, in some cases the thermodynamic properties are highlighted in the literature referenced in the main manuscript. Riedel et al. 2015 (Ref. 5) studied the role of the heat release on enhancing the diffusion of enzymes and mentioned urease to have an enthalpy of -59.6 kJ/mol (exothermic) on urea hydrolysis. Ilgen et al. 2017 (Ref. 9) commented on the enthalpy of aldolase which is estimated to range from 30 to 60 kJ/mol (endothermic) for fructose-1,6-bisphosphate catalysis. Additionally, in the “Enzyme Technology” book by Martin Chaplin and Christopher Bucke (Cambridge University Press, 1990) it can be found that β -D-glucose oxidation by glucose oxidase has an enthalpy of -80 kJ/mol (exothermic). Regarding acetylcholinesterase, the enthalpy of acetylcholine hydrolysis was found to be exothermic around -4 kJ/mol by Brown and Chattopadhyay 1985 and ranging from -5.43 to -38.02 kJ/mol by Draczkowski et al. 2015.

Given these reaction enthalpies no conclusion can be extracted in relation to the role of exothermicity/endothermicity on self-propulsion and further and detailed studies on these properties need to be addressed. We plan to study the effect of thermodynamics as one of our future projects, but initial tests of measuring the heat generated by urease micromotors show no increase in temperature. This could be due to a low sensitivity detection (0.5 °C) and still needs to be improved to extract any conclusion.

Comment 8: It's not obvious to me which symmetry is broken to achieve the directed motion? Do they have a patchy structure of the enzymes or a smooth uniform covering? If the latter, how was a monolayer coverage of the shell ensured?

Response 8: Yes, the referee is right in pointing these questions out. In our previous work Patino et al., JACS 2018, we reported a full characterization of enzyme distribution and number of immobilized enzymes onto a similar particle structure (the only difference was that in the previous work the polystyrene core was not removed as in this recent work), using Stochastically Optical Reconstruction Microscopy (STORM). A non-homogeneous distribution of enzymes on the micromotor surface was observed (Figure R8), which lead to think that in each particle there was a privileged direction in which the enzyme distribution symmetry is broken.

Figure R8. Analysis of 3D enzyme distribution on the micromotors surface. A, D) 3D reconstruction of single urease molecules detected by STORM. B, E) 3D density maps obtained by computational analysis of STORM imaging. C, F) Frequency of enzyme density detections per μm^2 . (Figure 3 from Patino et al., JACS 2018)

Comment 9: In Fig.2(c) why is the range of concentration of GLC so different to the other substrates?

Response 9: We selected a range of concentrations based on each enzyme kinetics specifications to ensure the exploration of the substrate concentrations that are higher enough than the K_M to achieve the maximum activity commonly known as the plateau of the Michaelis-Menten kinetics. The data is extracted from BRENDA, The Comprehensive Enzyme Information System (<https://www.brenda-enzymes.org/index.php>). Regarding glucose oxidase enzyme from *Aspergillus niger*, the K_M detected could reach around 100 mM glucose, so higher concentrations had to be explored.

Comment 10: To test the effect of conformational changes, why not study the differences that arise for different substrates of one enzyme? For example, the 2 substrates of aldolase as in Rago et. al 2015.

Response 10: We thank the reviewer for this valuable suggestion. We believe that using alternative substrates for the same enzyme and comparing their different effects on catalytic activity and conformational changes would definitely be interesting to explore in a follow-up paper. Indeed, the substrate at which the enzyme is exposed is decisive for both the catalytic turnover and the conformational changes that take place as shown by Rago et al. However, in the present manuscript, we decided to focus on the most active enzyme in terms of catalysis and self-propulsion (urease) and evaluate how different inhibitors (acetohydroxamic acid and

β -mercaptoethanol) might affect the flap conformational dynamics, modify enzymatic activity and regulate its micromotor motion.

Comment 11: If force is proportional to k_{cat} , compare UR-HSMM with 1 UR attached to the shell with ALS-HSMM with 1800 ALS attached, GOx-HSMM with 25 GOx attached and AChE-HSMM with 2 AChE; hence, two orders of magnitude difference between ALS and UR. However, Fig. 3 and Supplementary Fig. 13 and 14 do not show two orders of magnitude difference in the number of UR and ALS the silica shell is functionalized with.

Response 11: It would be interesting to explore the exact proportion of turnover number applied to the enzyme number of the 4 enzymes. Nevertheless, the turnover number per enzyme, not the overall activity of the particle, seems to be the crucial factor that modifies self-propulsion. This is clearly pointed out on Supplementary Figures 13 where the quantity of urease attached is increased but that does not result on an increased speed. Also, the experiment suggested by the reviewers offers difficulties on establishing an exact comparison since i) the exact number of enzymes attached cannot be controlled so accurately, ii) the multimeric conformation of each enzyme is different and iii) the proportions of each multimeric conformation for each enzyme is also different. Because of these reasons, we already address the issue of the proportionality of the propulsion force to the k_{cat} through the exposure of a reversible inhibitor and detecting the reduction of catalytic turnover of urease proportional to the experimental speed. This system seems more adequate since it affects the overall activity affecting the capacity of individual enzymes to catalyse, not just deleting or adding enzymes so the overall activity decreases or increases.

Comment 12: Under what conditions are the conformational changes measured in the MD simulations?

Response 12: The MD simulations used to evaluate the conformational changes have been performed in the NVT ensemble at 300K using water as an explicit solvent. To prepare each system for the MD simulations, we carried out the following steps. First, each enzyme was solvated in a pre-equilibrated truncated cuboid box with a 10 Å buffer of water molecules. Then, the system was neutralized by the addition of explicit counterions (Na^+ and Cl^-). Finally, the protocol described in the Molecular Dynamics Simulations section of the SI was followed to carry out the MD simulations at 300K in the NVT ensemble.

Comment 13: Is it possible that a higher flexibility was found in UR and AChE because they are bigger?

Response 13: Flexibility is determined by the amino acid sequence and structural properties of the enzyme scaffold. Therefore, it is not correlated with the size.

Comment 14: In reference to conformational changes, the suggestion that "In ALS and GOx, these changes are decoupled from catalysis and act unilaterally" and also the suggestion that

conformational changes in ALS and GOx have "no direct contribution to substrate binding and thus catalysis" is not justified, as it is in contradiction with Rago et. al. 2015.

Response 14: We thank referee 3 for this suggestion. We claim that the conformational changes of aldolase (ALS) do not contribute directly to catalysis of substrate as a closing gate precondition, but it may well still be possible that conformational changes play a role on specific substrate recognition and catalysis. We have changed the sentence in page 13, line 2 that now reads: "In ALS and GOx, these changes are far from the active site and have a minor effect along the catalytic cycle".

Comment 15: One final comment about giving credit: stochastic swimming has been discussed in an extensive literature, which is mentioned and expanded on in Ref. 6 in addition to discussing the collective heating mechanism that the manuscript currently mentions. It is not appropriate to give the credit about this concept to Ref. 7, as it is a review article and is discussing other people's works.

Response 15: We thank the reviewer for noticing this. First, we think that probably the reviewer meant "Ref. 6" when writing "Ref. 7" in the last phrase, since otherwise we do not understand why Ref. 6 would be mentioned before in the comment, specially taking into account that Ref. 6 is indeed a review while Ref. 7 is not. We have revised the main text, re-organized the references and added "discussed by Golestanian" in page 3, line 1, for Ref. 6, who in 2015 introduced the concept of "collective heating" for the first time, to the best of our knowledge, although being a review.

REVIEWERS' COMMENTS:

Reviewer #1 (Remarks to the Author):

The authors have put some effort into improving the manuscript. While I do not necessarily agree with some of the explanations the authors are providing, especially when it comes to the control of the activity of the immobilized enzymes. Nonetheless, I also think the manuscript is worth publishing now. It will certainly initiate discussions and contribute to the development of the swimmers.

Reviewer #2 (Remarks to the Author):

The authors' changes are satisfactory and I am happy to recommend this paper for publication in Nature Communications. The paper provides convincing evidence, based on four enzymes, that those enzymes with large conformational motions undergo larger catalysis-driven displacements, and that this driven motion is most likely in a specific direction in the frame of reference of the enzyme. These results are augmented by computational studies that support the experiments.

R. Dean Astumian

Reviewer #3 (Remarks to the Author):

The authors have addressed most of my comments in the revision, except for the two comments on credit, which might be due to misunderstanding. So, I provide some information to clarify my points:

Comment 2: Coating the sphere by one enzyme of many, but asymmetrically, conceptually belongs to the same category of phenomena. The difference in numbers is irrelevant if we are still using the same conceptual framework. I do find the response of the authors acceptable, when they say because of these differences they do not think these two descriptions belong to the same category and a credit should be given.

Comment 15: Ref. [7] Sakaue, T., Kapral, R. & Mikhailov, A. S. Nanoscale swimmers: Hydrodynamic interactions and propulsion of molecular machines. Eur. Phys. J. B 75, 381–387(2010), is a review paper that discussed the work done on stochastic swimming earlier in the following references:

Mechanical Response of a Small Swimmer Driven by Conformational Transitions, R. Golestanian and A. Ajdari, Phys. Rev. Lett. 100, 038101(22 January 2008).

Stochastic low Reynolds number swimmers, R. Golestanian and A. Ajdari, J. Phys.: Condens. Matter 21 204104 (2009)

The new contribution in Ref. [7] is a schematic figure of an enzyme to describe the stochastic swimmer rather than the idealized models of beads used in the original references.

REVIEWERS' COMMENTS:

Reviewer #1 (Remarks to the Author):

The authors have put some effort into improving the manuscript. While I do not necessarily agree with some of the explanations the authors are providing, especially when it comes to the control of the activity of the immobilized enzymes. Nonetheless, I also think the manuscript is worth publishing now. It will certainly initiate discussions and contribute to the development of the swimmers.

Response Reviewer #1: We are grateful to read that our effort is acknowledged and that our work is considered to trigger debate in the community to ultimately contribute to the optimization of micromotors.

Reviewer #2 (Remarks to the Author):

The authors' changes are satisfactory and I am happy to recommend this paper for publication in Nature Communications. The paper provides convincing evidence, based on four enzymes, that those enzymes with large conformational motions undergo larger catalysis-driven displacements, and that this driven motion is most likely in a specific direction in the frame of reference of the enzyme. These results are augmented by computational studies that support the experiments.

R. Dean Astumian

Response Reviewer #2: We appreciate that the Reviewer #2 is satisfied with the changes applied to answer his/her comments and that now this work is considered as suitable for publication in Nature Communications.

Reviewer #3 (Remarks to the Author):

The authors have addressed most of my comments in the revision, except for the two comments on credit, which might be due to misunderstanding. So, I provide some information to clarify my points:

Comment 2: Coating the sphere by one enzyme of many, but asymmetrically, conceptually belongs to the same category of phenomena. The difference in numbers is irrelevant if we are still using the same conceptual framework. I do find the response of the authors acceptable, when they say because of these differences they do not think these two descriptions belong to the same category and a credit should be given.

Response 2: We appreciate that the referee thinks our response and reasoning is acceptable.

Comment 15: Ref. [7] Sakaue, T., Kapral, R. & Mikhailov, A. S. Nanoscale swimmers: Hydrodynamic interactions and propulsion of molecular machines. *Eur. Phys. J. B* 75, 381–387(2010), is a review paper that discussed the work done on stochastic swimming earlier in the following references:

Mechanical Response of a Small Swimmer Driven by Conformational Transitions, R. Golestanian and A. Ajdari, *Phys. Rev. Lett.* 100, 038101(22 January 2008).

Stochastic low Reynolds number swimmers, R. Golestanian and A. Ajdari, *J. Phys.: Condens. Matter* 21 204104 (2009)

The new contribution in Ref. [7] is a schematic figure of an enzyme to describe the stochastic swimmer rather than the idealized models of beads used in the original references.

Response 15: We agree that giving credit to original work is more appropriate than to a review in the field, thus we have changed Ref. 7:

Sakaue, T., Kapral, R. & Mikhailov, A. S. Nanoscale swimmers: Hydrodynamic interactions and propulsion of molecular machines. Eur. Phys. J. B 75, 381–387(2010)

for

Mechanical Response of a Small Swimmer Driven by Conformational Transitions, R. Golestanian and A. Ajdari, Phys. Rev. Lett. 100, 038101(22 January 2008)

and

Stochastic low Reynolds number swimmers, R. Golestanian and A. Ajdari, J. Phys.: Condens. Matter 21 204104 (2009)